# Selecting the independent coordinates of manifolds with large aspect ratios

**Yu-Chia Chen**
Department of Electrical & Computer Engineering
University of Washington
Seattle, WA 98195
yuchaz@uw.edu

**Marina Meilă**
Department of Statistics
University of Washington
Seattle, WA 98195
mmp2@uw.edu

## Abstract

Many manifold embedding algorithms fail apparently when the data manifold has a large aspect ratio (such as a long, thin strip). Here, we formulate success and failure in terms of finding a smooth embedding, showing also that the problem is pervasive and more complex than previously recognized. Mathematically, success is possible under very broad conditions, provided that embedding is done by carefully selected eigenfunctions of the Laplace-Beltrami operator $\Delta_{\mathcal{M}}$. Hence, we propose a bicriterial Independent Eigencoordinate Selection (IES) algorithm that selects smooth embeddings with few eigenvectors. The algorithm is grounded in theory, has low computational overhead, and is successful on synthetic and large real data.

## 1   Motivation

We study a well-documented deficiency of manifold learning algorithms. Namely, as shown in [GZKR08], algorithms such as Laplacian Eigenmaps (LE), Local Tangent Space Alignment (LTSA), Hessian Eigenmaps (HLLE), and Diffusion Maps (DM) fail spectacularly when the data has a large aspect ratio, that is, it extends much more in one geodesic direction than in others. This problem, illustrated by the strip in Figure 1, was studied in [GZKR08] from a *linear algebraic* perspective; [GZKR08] show that, especially when noise is present, the problem is pervasive.

In the present paper, we revisit the problem from a *differential geometric* perspective. First, we define failure not as distortion, but as drop in the *rank* of the mapping $\phi$ represented by the embedding algorithm. In other words, the algorithm fails when the map $\phi$ is not invertible, or, equivalently, when the dimension $\dim \phi(\mathcal{M}) < \dim \mathcal{M} = d$, where $\mathcal{M}$ represents the idealized data manifold, and $\dim$ denotes the intrinsic dimension. Figure 1 demonstrates that the problem is fixed by choosing the eigenvectors with care. We call this problem the *Independent Eigencoordinate Selection* (IES) problem, formulate it and explain its challenges in Section 3.

Our second main contribution (Section 4) is to design a bicriterial method that will select from a set of *coordinate functions* $\phi_1, \dots \phi_m$, a subset $S$ of small size that provides a smooth full-dimensional embedding of the data. The IES problem requires searching over a combinatorial number of sets. We show (Section 4) how to drastically reduce the computational burden per set for our algorithm. Third, we analyze the proposed criterion under asymptotic limit (Section 5). Finally (Section 6), we show examples of successful selection on real and synthetic data. The experiments also demonstrate that users of manifold learning for other than toy data *must* be aware of the IES problem and have tools for handling it. Notations table, proofs, a library of hard examples, extra experiments and analyses are in Supplements A–H; Figure/Table/Equation references with prefix S are in the Supplement.

## 2 Background on manifold learning

**Manifold learning (ML) and intrinsic geometry** Suppose we observe data $\mathbf{X} \in \mathbb{R}^{n \times D}$, with data points denoted by $\mathbf{x}_i \in \mathbb{R}^D \; \forall \; i \in [n]$, that are sampled from a *smooth*[1] $d$-dimensional submanifold $\mathcal{M} \subset \mathbb{R}^D$. Manifold Learning algorithms map $\mathbf{x}_i, i \in [n]$ to $\mathbf{y}_i = \phi(\mathbf{x}_i) \in \mathbb{R}^s$, where $d \leq s \ll D$, thus reducing the dimension of the data $\mathbf{X}$ while preserving (some of) its properties. Here we present the LE/DM algorithm, but our results can be applied to other ML methods with slight modification. The DM [CL06, NLCK06] algorithm embeds the data by solving the minimum eigen-problem of the *renormalized graph Laplacian* [CL06] matrix $\mathbf{L}$. The desired $m$ dimensional embedding coordinates are obtained from the second to $m + 1$-th principal eigenvectors of graph Laplacian $\mathbf{L}$, with $0 = \lambda_0 < \lambda_1 \leq \ldots \leq \lambda_m$, i.e., $\mathbf{y}_i = (\phi_1(\mathbf{x}_i), \ldots \phi_m(\mathbf{x}_i))$ (see also Supplement B).

To analyze ML algorithms, it is useful to consider the limit of the mapping $\phi$ when the data is the entire manifold $\mathcal{M}$. We denote this limit also by $\phi$, and its image by $\phi(\mathcal{M}) \in \mathbb{R}^m$. For standard algorithms such as LE/DM, it is known that this limit exists [CL06, BN07, HAvL05, HAvL07, THJ10]. One of the fundamental requirements of ML is to preserve the neighborhood relations in the original data. In mathematical terms, we require that $\phi : \mathcal{M} \to \phi(\mathcal{M})$ is a *smooth embedding*, i.e., that $\phi$ is a smooth function (i.e. does not break existing neighborhood relations) whose Jacobian $\mathbf{D}\phi(\mathbf{x})$ is full rank $d$ at each $\mathbf{x} \in \mathcal{M}$ (i.e. does not create new neighborhood relations).

**The pushforward Riemannian metric** A smooth $\phi$ does not typically preserve geometric quantities such as distances along curves in $\mathcal{M}$. These concepts are captured by *Riemannian geometry*, and we additionally assume that $(\mathcal{M}, g)$ is a *Riemannian manifold*, with the metric $g$ induced from $\mathbb{R}^D$. One can always associate with $\phi(\mathcal{M})$ a Riemannian metric $g_{*\phi}$, called the *pushforward Riemannian metric* [Lee03], which preserves the geometry of $(\mathcal{M}, g)$; $g_{*\phi}$ is defined by

$$\langle \mathbf{u}, \mathbf{v} \rangle_{g_{*\phi}(\mathbf{x})} = \left\langle \mathbf{D}\phi^{-1}(\mathbf{x})\mathbf{u}, \mathbf{D}\phi^{-1}(\mathbf{x})\mathbf{v} \right\rangle_{g(\mathbf{x})} \text{ for all } \mathbf{u}, \mathbf{v} \in \mathcal{T}_{\phi(\mathbf{x})}\phi(\mathcal{M}) \tag{1}$$

In the above, $\mathcal{T}_{\mathbf{x}}\mathcal{M}$, $\mathcal{T}_{\phi(\mathbf{x})}\phi(\mathcal{M})$ are tangent subspaces, $\mathbf{D}\phi^{-1}(\mathbf{x})$ maps vectors from $\mathcal{T}_{\phi(\mathbf{x})}\phi(\mathcal{M})$ to $\mathcal{T}_{\mathbf{x}}\mathcal{M}$, and $\langle , \rangle$ is the Euclidean scalar product. For each $\phi(\mathbf{x}_i)$, the associated pushforward Riemannian metric expressed in the coordinates of $\mathbb{R}^m$, is a symmetric, semi-positive definite $m \times m$ matrix $\mathbf{G}(i)$ of rank $d$. The scalar product $\langle \mathbf{u}, \mathbf{v} \rangle_{g_{*\phi}(\mathbf{x}_i)}$ takes the form $\mathbf{u}^\top \mathbf{G}(i)\mathbf{v}$. Given an embedding $\mathbf{Y} = \phi(\mathbf{X})$, $\mathbf{G}(i)$ can be estimated by Algorithm 1 (RMETRIC) of [PM13]. The RMETRIC also returns the *co-metric* $\mathbf{H}(i)$, which is the pseudo-inverse of the metric $\mathbf{G}(i)$, and its Singular Value

---

**Algorithm 1:** RMETRIC

**Input** : Embedding $\mathbf{Y} \in \mathbb{R}^{n \times m}$, Laplacian $\mathbf{L}$, intrinsic dimension $d$

**1** **for** *all* $\mathbf{y}_i \in \mathbf{Y}, k = 1 \to m, l = 1 \to m$ **do**

**2** $\quad [\tilde{\mathbf{H}}(i)]_{kl} = \sum_{j \neq i} L_{ij}(y_{jl} - y_{il})(y_{jk} - y_{ik})$

**3** **end**

**4** **for** $i = 1 \to n$ **do**

**5** $\quad \mathbf{U}(i), \mathbf{\Sigma}(i) \leftarrow \text{REDUCEDRANKSVD}(\tilde{\mathbf{H}}(i), d)$

**6** $\quad \mathbf{H}(i) = \mathbf{U}(i)\mathbf{\Sigma}(i)\mathbf{U}(i)^\top$

**7** $\quad \mathbf{G}(i) = \mathbf{U}(i)\mathbf{\Sigma}^{-1}(i)\mathbf{U}(i)^\top$

**8** **end**

**Return:** $\mathbf{G}(i), \mathbf{H}(i) \in \mathbb{R}^{m \times m}, \mathbf{U}(i) \in \mathbb{R}^{m \times d}$, $\mathbf{\Sigma}(i) \in \mathbb{R}^{d \times d}$, for $i \in [n]$

---

Decomposition $\mathbf{\Sigma}(i), \mathbf{U}(i) \in \mathbb{R}^{m \times d}$. The latter represents an orthogonal basis of $\mathcal{T}_{\phi(\mathbf{x})}(\phi(\mathcal{M}))$.

## 3 IES problem, related work, and challenges

**An example** Consider a continuous two dimensional strip with width $W$, height $H$, and *aspect ratio* $W/H \geq 1$, parametrized by coordinates $w \in [0, W], h \in [0, H]$. The eigenvalues and eigenfunctions of the Laplace-Beltrami operator $\Delta$ with von Neumann boundary conditions [Str07] are $\lambda_{k_1, k_2} = \left(\frac{k_1 \pi}{W}\right)^2 + \left(\frac{k_2 \pi}{H}\right)^2$, respectively $\phi_{k_1, k_2}(w, h) = \cos\left(\frac{k_1 \pi w}{W}\right) \cos\left(\frac{k_2 \pi h}{H}\right)$. Eigenfunctions $\phi_{1,0}, \phi_{0,1}$ are in bijection with the $w, h$ coordinates (and give a full rank embedding), while the mapping by $\phi_{1,0}, \phi_{2,0}$ provides no extra information regarding the second dimension $h$ in the underlying manifold (and is rank 1). Theoretically, one can choose as coordinates eigenfunctions indexed by $(k_1, 0), (0, k_2)$, but, in practice, $k_1$, and $k_2$ are usually

unknown, as the eigenvalues are index by their rank $0 = \lambda_0 < \lambda_1 \leq \lambda_2 \leq \cdots$. For a two dimensional strip, it is known [Str07] that $\lambda_{1,0}$ always corresponds to $\lambda_1$ and $\lambda_{0,1}$ corresponds to $\lambda_{(\lceil W/H \rceil)}$. Therefore, when $W/H > 2$, the mapping of the strip to $\mathbb{R}^2$ by $\phi_1, \phi_2$ is low rank, while the mapping by $\phi_1, \phi_{\lceil W/H \rceil}$ is full rank. Note that other mappings of rank 2 exist, e.g., $\phi_1, \phi_{\lceil W/H \rceil + 2}$ ($k_1 = k_2 = 1$ in Figure 1b). These embeddings reflect progressively higher frequencies, as the corresponding eigenvalues grow larger.

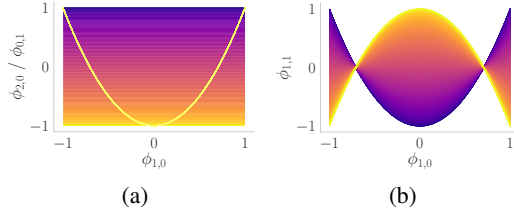

(a)             (b)

Figure 1: (a) Eigenfunction $\phi_{1,0}$ versus $\phi_{2,0}$ (curve) or $\phi_{0,1}$ (two dimensional manifold). (b) Eigenfunction $\phi_{1,0}$ versus $\phi_{1,1}$. All three manifolds are colored by the parameterization $h$.

**Prior work** [GZKR08] is the first work to give the IES problem a rigurous analysis. Their paper focuses on rectangles, and the failure illustrated in Figure 1a is defined as obtaining a mapping $\mathbf{Y} = \phi(\mathbf{X})$ that is not *affinely equivalent* with the original data. They call this the *Price of Normalization* and explain it in terms of the variances along $w$ and $h$. [DTCK18] is the first to frame the failure in terms of the rank of $\phi_S = \{\phi_k : k \in S \subseteq [m]\}$, calling it the *repeated eigendirection problem*. They propose a heuristic, LLRCOORDSEARCH, based on the observation that if $\phi_k$ is a repeated eigendirection of $\phi_1, \cdots, \phi_{k-1}$, one can fit $\phi_k$ with *local linear regression* on predictors $\phi_{[k-1]}$ with low leave-one-out errors $r_k$. A sequential algorithm [BM17] with an unpredictability constraint in the eigenproblem has also been proposed. Under their framework, the $k$-th coordinate $\phi_k$ is obtained from the top eigenvector of the modified kernel matrix $\tilde{\mathbf{K}}_k$, which is constructed by the original kernel $\mathbf{K}$ and $\phi_1, \cdots, \phi_{k-1}$.

**Existence of solution** Before trying to find an algorithmic solution to the IES problem, we ask the question whether this is even possible, in the smooth manifold setting. Positive answers are given in [Por16], which proves that isometric embeddings by DM with finite $m$ are possible, and more recently in [Bat14], which proves that any closed, connected Riemannian manifold $\mathcal{M}$ can be smoothly embedded by its Laplacian eigenfunctions $\phi_{[m]}$ into $\mathbb{R}^m$ for some $m$, which depends only on the intrinsic dimension $d$ of $\mathcal{M}$, the volume of $\mathcal{M}$, and lower bounds for *injectivity radius* and *Ricci curvature*. The example in Figure 1a demonstrates that, typically, not all $m$ eigenfunctions are needed. I.e., there exists a set $S \subset [m]$, so that $\phi_S$ is also a smooth embedding. We follow [DTCK18] in calling such a set $S$ *independent*. It is not known how to find an independent $S$ analytically for a given $\mathcal{M}$, except in special cases such as the strip. In this paper, we propose a *finite sample* and algorithmic solution, and we support it with asymptotic theoretical analysis.

**The IES Problem** We are given data $\mathbf{X}$, and the output of an embedding algorithm (DM for simplicity) $\mathbf{Y} = \phi(\mathbf{X}) = [\phi_1, \cdots, \phi_m] \in \mathbb{R}^{n \times m}$. We assume that $\mathbf{X}$ is sampled from a $d$-dimensional manifold $\mathcal{M}$, with known $d$, and that $m$ is sufficiently large so that $\phi(\mathcal{M})$ is a smooth embedding. Further, we assume that there is a set $S \subseteq [m]$, with $|S| = s \leq m$, so that $\phi_S$ is also a smooth embedding of $\mathcal{M}$. We propose to find such set $S$ so that the rank of $\phi_S$ is $d$ on $\mathcal{M}$ and $\phi_S$ varies as slowly as possible.

**Challenges** (1) Numerically, and on a finite sample, distiguishing between a full rank mapping and a rank-defective one is imprecise. Therefore, we substitute for rank the volume of a unit parallelogram in $\mathcal{T}_{\phi(\mathbf{x}_i)}\phi(\mathcal{M})$. (2) Since $\phi$ is *not* an isometry, we must separate the local distortions introduced by $\phi$ from the estimated rank of $\phi$ at $\mathbf{x}$. (3) Finding the optimal balance between the above desired properties. (4) In [Bat14] it is strongly suggested that $s$ the number of eigenfunctions needed may exceed the *Whitney embedding dimension* ($\leq 2d$), and that this number may depend on injectivity radius, aspect ratio, and so on. Supplement G shows an example of a flat 2-manifold, the *strip with cavity*, for which $s > 2$. In this paper, we assume that $s$ and $m$ are given and focus on selecting $S$ with $|S| = s$; for completeness, in Supplement G we present a heuristic to select $s$.

**(Global) functional dependencies, knots and crossings** Before we proceed, we describe three different ways a mapping $\phi(\mathcal{M})$ can fail to be invertible. The first, *(global) functional dependency* is the case when rank $\mathbf{D}\phi < d$ on an open subset of $\mathcal{M}$, or on all of $\mathcal{M}$ (yellow curve in Figure 1a); this is the case most widely recognized in the literature (e.g., [GZKR08, DTCK18]). The *knot* is the case when rank $\mathbf{D}\phi < d$ at an isolated point (Figure 1b). Third, the *crossing* (Figure S8 in

Supplement H) is the case when $\phi : \mathcal{M} \to \phi(\mathcal{M})$ is not invertible at $\mathbf{x}$, but $\mathcal{M}$ can be covered with open sets $U$ such that the restriction $\phi : U \to \phi(U)$ has full rank $d$. Combinations of these three exemplary cases can occur. The criteria and approach we define are based on the (surrogate) rank of $\phi$, therefore they will not rule out all crossings. We leave the problem of crossings in manifold embeddings to future work, as we believe that it requires an entirely separate approach (based, e.g., or the injectivity radius or density in the co-tangent bundle rather than differential structure).

## 4 Criteria and algorithm

**A geometric criterion**  We start with the main idea in evaluating the quality of a subset $S$ of coordinate functions. At each data point $i$, we consider the orthogonal basis $\mathbf{U}(i) \in \mathbb{R}^{m \times d}$ of the $d$ dimensional tangent subspace $\mathcal{T}_{\phi(\mathbf{x}_i)}\phi(\mathcal{M})$. The projection of the columns of $\mathbf{U}(i)$ onto the subspace $\mathcal{T}_{\phi(\mathbf{x}_i)}\phi_S(\mathcal{M})$ is $\mathbf{U}(i)[S,:] \equiv \mathbf{U}_S(i)$. The following Lemma connects $\mathbf{U}_S(i)$ and the co-metric $\mathbf{H}_S(i)$ defined by $\phi_S$, with the *full* $\mathbf{H}(i)$.

**Lemma 1.** *Let* $\mathbf{H}(i) = \mathbf{U}(i)\mathbf{\Sigma}(i)\mathbf{U}(i)^\top$ *be the co-metric defined by embedding* $\phi$, $S \subseteq [m]$, $\mathbf{H}_S(i)$ *and* $\mathbf{U}_S(i)$ *defined above. Then* $\mathbf{H}_S(i) = \mathbf{U}_S(i)\mathbf{\Sigma}(i)\mathbf{U}_S(i)^\top = \mathbf{H}(i)[S,S]$.

The proof is straightforward and left to the reader. Note that Lemma 1 is responsible for the efficiency of the search over sets $S$, given that the push-forward co-metric $\mathbf{H}_S$ can be readily obtained as a submatrix of $\mathbf{H}$. Denote by $\mathbf{u}_k^S(i)$ the $k$-th column of $\mathbf{U}_S(i)$. We further normalize each $\mathbf{u}_k^S$ to length 1 and define the *normalized projected volume* $\mathrm{Vol}_{\mathrm{norm}}(S,i) = \frac{\sqrt{\det(\mathbf{U}_S(i)^\top \mathbf{U}_S(i))}}{\prod_{k=1}^d \|\mathbf{u}_k^S(i)\|_2}$. Conceptually, $\mathrm{Vol}_{\mathrm{norm}}(S,i)$ is the volume spanned by a (non-orthonormal) "basis" of unit vectors in $\mathcal{T}_{\phi_S(\mathbf{x}_i)}\phi_S(\mathcal{M})$; $\mathrm{Vol}_{\mathrm{norm}}(S,i) = 1$ when $\mathbf{U}_S(i)$ is orthogonal, and it is 0 when $\mathrm{rank}\,\mathbf{H}_S(i) < d$. In Figure 1a, the $\mathrm{Vol}_{\mathrm{norm}}(\{1,2\})$ with $\phi_{\{1,2\}} = \{\phi_{1,0}, \phi_{2,0}\}$ is close to zero, since the projection of the two tangent vectors is parallel to the yellow curve; however $\mathrm{Vol}_{\mathrm{norm}}(\{1, \lceil w/h \rceil\}, i)$ is almost 1, because the projections of the tangent vectors $\mathbf{U}(i)$ will be (approximately) orthogonal. Hence, $\mathrm{Vol}_{\mathrm{norm}}(S,i)$ away from 0 indicates a non-singular $\phi_S$ at $i$, and we use the average $\log \mathrm{Vol}_{\mathrm{norm}}(S,i)$, which penalizes values near 0 highly, as the *rank quality* $\mathfrak{R}(S)$ of $S$.

Higher frequency $\phi_S$ maps with high $\mathfrak{R}(S)$ may exist, being either smooth, such as the embeddings of the strip mentioned previously, or containing knots involving only small fraction of points, such as $\phi_{\phi_{1,0},\phi_{1,1}}$ in Figure 1a. To choose the lowest frequency, slowest varying smooth map, a regularization term consisting of the eigenvalues $\lambda_k$, $k \in S$, of the graph Laplacian $\mathbf{L}$ is added, obtaining the criterion

$$\mathfrak{L}(S;\zeta) = \underbrace{\frac{1}{n}\sum_{i=1}^n \log \sqrt{\det\left(\mathbf{U}_S(i)^\top \mathbf{U}_S(i)\right)}}_{\mathfrak{R}_1(S) = \frac{1}{n}\sum_{i=1}^n \mathfrak{R}_1(S;i)} - \underbrace{\frac{1}{n}\sum_{i=1}^n \sum_{k=1}^d \log \|\mathbf{u}_k^S(i)\|_2}_{\mathfrak{R}_2(S) = \frac{1}{n}\sum_{i=1}^n \mathfrak{R}_2(S;i)} - \zeta \sum_{k \in S} \lambda_k \quad (2)$$

**Search algorithm**  With this criterion, the IES problem turns into a subset selection problem parametrized by $\zeta$

$$S_*(\zeta) = \underset{S \subseteq [m]; |S|=s, 1 \in S}{\mathrm{argmax}} \mathfrak{L}(S;\zeta) \quad (3)$$

Note that we force the first coordinate $\phi_1$ to always be chosen, since this coordinate cannot be functionally dependent on previous ones, and, in the case of DM, it also has lowest frequency. Note also that $\mathfrak{R}_1$ and $\mathfrak{R}_2$ are both submodular set function (proof in Supplement C.3). For large $s$ and $d$, algorithms for optimizing over the difference of submodular functions can be used (e.g., see [IB12]). For the experiments in this paper, we have $m = 20$ and

---

**Algorithm 2:** IndEigenSearch

**Input** : Data $\mathbf{X}$, bandwith $\varepsilon$, intrinsic dimension $d$, embedding dimension $s$, regularizer $\zeta$

1   $\mathbf{Y} \in \mathbb{R}^{n \times m}, \mathbf{L}, \boldsymbol{\lambda} \in \mathbb{R}^m \leftarrow \text{DiffMap}(\mathbf{X}, \varepsilon)$
2   $\mathbf{U}(i), \cdots, \mathbf{U}(n) \leftarrow \text{RMetric}(\mathbf{Y}, \mathbf{L}, d)$
3   **for** $S \in \{S' \subseteq [m] : |S'| = s, 1 \in S'\}$ **do**
4     $\mathfrak{R}_1(S) \leftarrow 0; \mathfrak{R}_2(S) \leftarrow 0$
5     **for** $i = 1, \cdots, n$ **do**
6       $\mathbf{U}_S(i) \leftarrow \mathbf{U}(i)[S,:]$
7       $\mathfrak{R}_1(S) \mathrel{+}= \frac{1}{2n} \cdot \log \det \left(\mathbf{U}_S(i)^\top \mathbf{U}_S(i)\right)$
8       $\mathfrak{R}_2(S) \mathrel{+}= \frac{1}{n} \cdot \sum_{k=1}^d \log \|u_k^S(i)\|_2$
9     **end**
10    $\mathfrak{L}(S;\zeta) = \mathfrak{R}_1(S) - \mathfrak{R}_2(S) - \zeta \sum_{k \in S} \lambda_k$
11 **end**
12 $S_* = \mathrm{argmax}_S \mathfrak{L}(S;\zeta)$
**Return:** Independent eigencoordinates set $S_*$

$d, s = 2 \sim 4$, which enables us to use exhaustive search to handle (3). The exact search algorithm is summarized in Algorithm 2 INDEIGENSEARCH. A greedy variant is also proposed and analyzed in Supplement D. Note that one might be able to search in the continuous space of all $s$-projections. We conjecture the objective function (2) will be a difference of convex function and leave the details as future work[2].

**Regularization path and choosing $\zeta$**   According to (2), the optimal subset $S_*$ depends on the parameter $\zeta$. The regularization path $\ell(\zeta) = \max_{S \subseteq [m]; |S|=s; 1 \in S} \mathfrak{L}(S; \zeta)$ is the upper envelope of multiple lines (each correspond to a set $S$) with slopes $-\sum_{k \in S} \lambda_k$ and intercepts $\mathfrak{R}(S)$. The larger $\zeta$ is, the more the lower frequency subset penalty prevails, and for sufficiently large $\zeta$ the algorithm will output $[s]$. In the supervised learning framework, the regularization parameters are often chosen by cross validation. Here we propose a second criterion, that effectively limits how much $\mathfrak{R}(S)$ may be ignored, or alternatively, bounds $\zeta$ by a data dependent quantity. Define the *leave-one-out regret* of point $i$ as follows

$$\mathfrak{D}(S, i) = \mathfrak{R}(S_*^i; [n] \backslash \{i\}) - \mathfrak{R}(S; [n] \backslash \{i\}) \text{ with } S_*^i = \text{argmax}_{S \subseteq [m]; |S|=s; 1 \in S} \mathfrak{R}(S; i) \quad (4)$$

In the above, we denote $\mathfrak{R}(S; T) = \frac{1}{|T|} \sum_{i \in T} \mathfrak{R}_1(S; i) - \mathfrak{R}_2(S; i)$ for some subset $T \subseteq [n]$. The quantity $\mathfrak{D}(S, i)$ in (4) measures the gain in $\mathfrak{R}$ if all the other points $[n] \backslash \{i\}$ choose the optimal subset $S_*^i$. If the regret $\mathfrak{D}(S, i)$ is larger than zero, it indicates that the alternative choice might be better compared to original choice $S$. Note that the mean value for all $i$, i.e., $\frac{1}{n} \sum_i \mathfrak{D}(S, i)$ depends also on the variability of the optimal choice of points $i$, $S_*^i$. Therefore, it might not favor an $S$, if $S$ is optimal for every $i \in [n]$. Instead, we propose to inspect the distribution of $\mathfrak{D}(S, i)$, and remove the sets $S$ for which $\alpha$'s percentile are larger than zero, e.g., $\alpha = 75\%$, recursively from $\zeta = \infty$ in decreasing order. Namely, the chosen set is $S_* = S_*(\zeta')$ with $\zeta' = \max_{\zeta \geq 0} \text{PERCENTILE}(\{\mathfrak{D}(S_*(\zeta), i)\}_{i=1}^n, \alpha) \leq 0$. The optimal $\zeta_*$ value is simply chosen to be the midpoint of all the $\zeta$'s that outputs set $S_*$ i.e., $\zeta_* = \frac{1}{2}(\zeta' + \zeta'')$, where $\zeta'' = \min_{\zeta \geq 0} S_*(\zeta) = S_*(\zeta')$. The procedure REGUPARAMSEARCH is summarized in Algorithm S5.

# 5   $\mathfrak{R}$ as Kullbach-Leibler divergence

In this section we analyze $\mathfrak{R}$ in its population version, and show that it is reminiscent of a Kullbach-Leibler divergence between *unnormalized* measures on $\phi_S(\mathcal{M})$. The population version of the regularization term takes the form of a well-known *smoothness* penalty on the embedding coordinates $\phi_S$. Proofs of the theorems can be found in Supplement C.

**Volume element and the Riemannian metric**   Consider a Riemannian manifold $(\mathcal{M}, g)$ mapped by a smooth embedding $\phi_S$ into $(\phi_S(\mathcal{M}), g_{*\phi_S})$, $\phi_S : \mathcal{M} \to \mathbb{R}^s$, where $g_{*\phi_S}$ is the *push-forward* metric defined in (1). A Riemannian metric $g$ induces a *Riemannian measure* on $\mathcal{M}$, with volume element $\sqrt{\det g}$. Denote now by $\mu_\mathcal{M}$, respectively $\mu_{\phi_S(\mathcal{M})}$ the Riemannian measures corresponding to the metrics induced on $\mathcal{M}, \phi_S(\mathcal{M})$ by the ambient spaces $\mathbb{R}^D, \mathbb{R}^s$; let $g$ be the former metric.

**Lemma 2.** *Let $S, \phi, \phi_S, \mathbf{H}_S(\mathbf{x}), \mathbf{U}_S(\mathbf{x}), \mathbf{\Sigma}(\mathbf{x})$ be defined as in Section 4 and Lemma 1. For simplicity, we denote by $\mathbf{H}_S(\mathbf{y}) \equiv \mathbf{H}_S(\phi_S^{-1}(\mathbf{y}))$, and similarly for $\mathbf{U}_S(\mathbf{y}), \mathbf{\Sigma}(\mathbf{y})$. Assume that $\phi_S$ is a smooth embedding. Then, for any measurable function $f : \mathcal{M} \to \mathbb{R}$,*

$$\int_\mathcal{M} f(\mathbf{x}) d\mu_\mathcal{M}(\mathbf{x}) = \int_{\phi_S(\mathcal{M})} f(\phi_S^{-1}(\mathbf{y})) j_S(y) d\mu_{\phi_S(\mathcal{M})}(\mathbf{y}), \quad (5)$$

*with*

$$j_S(\mathbf{y}) = 1/\text{Vol}(\mathbf{U}_S(\mathbf{y}) \mathbf{\Sigma}_S^{1/2}(\mathbf{y})). \quad (6)$$

**Asymptotic limit of $\mathfrak{R}$**   We now study the first term of our criterion in the limit of infinite sample size. We make the following assumptions.

**Assumption 1.** *The manifold $\mathcal{M}$ is compact of class $\mathcal{C}^3$, and there exists a set $S$, with $|S| = s$ so that $\phi_S$ is a smooth embedding of $\mathcal{M}$ in $\mathbb{R}^s$.*

**Assumption 2.** *The data are sampled from a distribution on $\mathcal{M}$ continuous with respect to $\mu_{\mathcal{M}}$, whose density is denoted by $p$.*

**Assumption 3.** *The estimate of $\mathbf{H}_S$ in Algorithm 1 computed w.r.t. the embedding $\phi_S$ is consistent.*

We know from [Bat14] that Assumption 1 is satisfied for the DM/LE embedding. The remaining assumptions are minimal requirements ensuring that limits of our quantities exist. Now consider the setting in Sections 3, in which we have a larger set of eigenfunctions, $\phi_{[m]}$ so that $[m]$ contains the set $S$ of Assumption 1. Denote by $\tilde{j}_S(\mathbf{y}) = \prod_{k=1}^{d} \left( \|\mathbf{u}_k^S(\mathbf{y})\| \sigma_k(\mathbf{y}))^{1/2} \right)^{-1}$ a new volume element, here $\sigma_k = [\boldsymbol{\Sigma}]_{kk}$.

**Theorem 3** (Limit of $\mathfrak{R}$). *Under Assumptions 1–3,*

$$\lim_{n \to \infty} \frac{1}{n} \sum_i \ln \mathfrak{R}(S, \mathbf{x}_i) = \mathfrak{R}(S, \mathcal{M}), \tag{7}$$

*and*

$$\mathfrak{R}(S, \mathcal{M}) = -\int_{\phi_S(\mathcal{M})} \ln \frac{j_S(\mathbf{y})}{\tilde{j}_S(\mathbf{y})} j_S(\mathbf{y}) p(\phi_S^{-1}(\mathbf{y})) d\mu_{\phi_S(\mathcal{M})}(\mathbf{y}) \stackrel{def}{=} -D(pj_S \| p\tilde{j}_S) \tag{8}$$

The expression $D(\cdot \| \cdot)$ represents a Kullbach-Leibler divergence. Note that $j_S \geq \tilde{j}_S$, which implies that $D$ is always positive, and that the measures defined by $pj_S, p\tilde{j}_S$ normalize to different values. By definition, local injectivity is related to the volume element $j$. Intuitively, $pj_S$ is the *observation* and $p\tilde{j}_S$, where $\tilde{j}_S$ is the minimum attainable for $j_S$, is the *model*; the objective itself is looking for a view $S$ of the data that agrees with the model.

It is known that $\lambda_k$, the $k$-th eigenvalue of the Laplacian, converges under certain technical conditions [BN07] to an eigenvalue of the Laplace-Beltrami operator $\Delta_{\mathcal{M}}$ and that

$$\lambda_k(\Delta_{\mathcal{M}}) = \langle \phi_k, \Delta_{\mathcal{M}} \phi_k \rangle = \int_{\mathcal{M}} \| \operatorname{grad} \phi_k(\mathbf{x}) \|_2^2 d\mu(\mathcal{M}). \tag{9}$$

Hence, a smaller value for the regularization term encourages the use of slow varying coordinate functions, as measured by the squared norm of their gradients, as in equation (9). Hence, under Assumptions 1, 2, 3, $\mathfrak{L}$ converges to

$$\mathfrak{L}(S, \mathcal{M}) = -D(pj_S \| p\tilde{j}_S) - \left( \frac{\zeta}{\lambda_1(\mathcal{M})} \right) \sum_{k \in S} \lambda_k(\mathcal{M}). \tag{10}$$

Since eigenvalues scale with the volume of $\mathcal{M}$, the rescaling of $\zeta$ in comparison with equation (2) makes the $\zeta$ above adimensional.

## 6 Experiments

We demonstrate the proposed algorithm on three synthetic datasets, one where the minimum embedding dimension $s$ equals $d$ ($\mathcal{D}_1$ *long strip*), and two ($\mathcal{D}_7$ *high torus* and $\mathcal{D}_{13}$ *three torus*) where $s > d$. The complete list of synthetic manifolds (transformations of 2 dimensional strips, 3 dimensional cubes, two and three tori, etc.) investigated can be found in Supplement H and Table S2. The examples have (i) aspect ratio of at least 4 (ii) points sampled *non-uniformly* from the underlying manifold $\mathcal{M}$, and (iii) Gaussian noise added. The sample size of the synthetic datasets is $n = 10,000$ unless otherwise stated. Additionally, we analyze several real datasets from chemistry and astronomy. All embeddings are computed with the DM algorithm, which outputs $m = 20$ eigenvectors. Hence, we examine 171 sets for $s = 3$ and 969 sets for $s = 4$. No more than 2 to 5 of these sets appear on the regularization path. Detailed experimental results are in Table S3. In this section, we show the original dataset $\mathbf{X}$, the embedding $\phi_{S_*}$, with $S_*$ selected by INDEIGENSEARCH and $\zeta_*$ from REGUPARAMSEARCH, and the maximizer sets on the regularization path with box plots of $\mathfrak{D}(S, i)$ as discussed in Section 4. The $\alpha$ threshold for REGUPARAMSEARCH is set to $75\%$. The kernel bandwidth $\varepsilon$ for synthetic datasets is chosen manually. For real datasets, $\varepsilon$ is optimized as in [JMM17]. All the experiments are replicated for more than 5 times, and the outputs are similar because of the large sample size $n$.

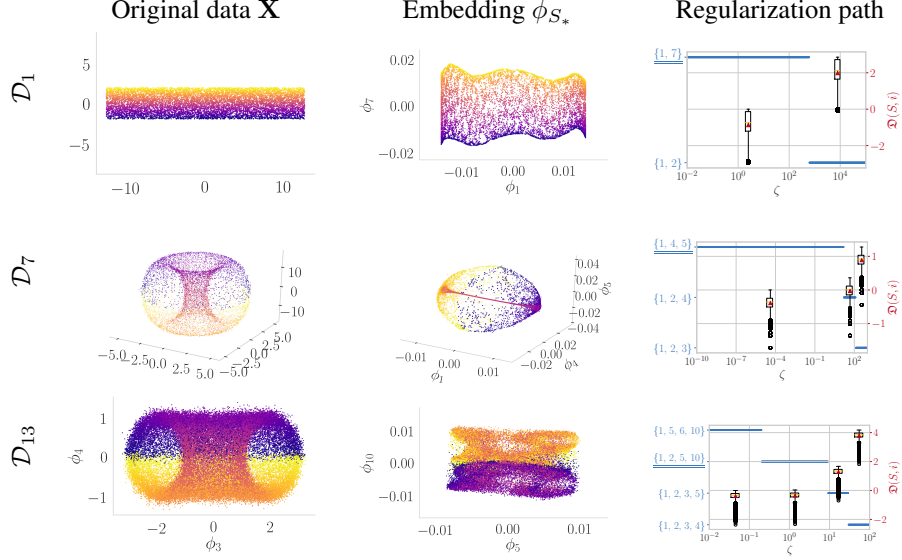

Figure 2: Experimental result for synthetic datasets. Rows correspond to different synthetic datasets (please refer to Table S2). Optimal subset $S_*$ is selected by INDEIGENSEARCH.

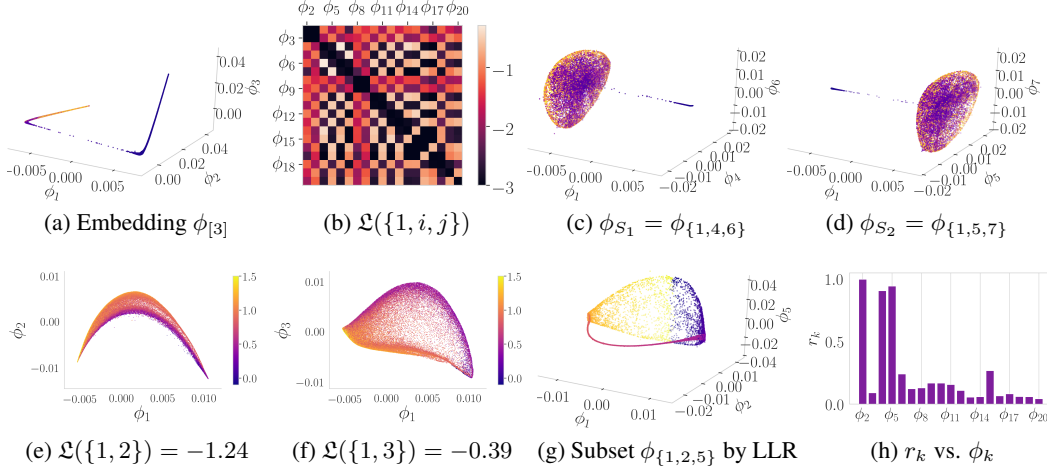

(a) Embedding $\phi_{[3]}$ (b) $\mathfrak{L}(\{1, i, j\})$ (c) $\phi_{S_1} = \phi_{\{1,4,6\}}$ (d) $\phi_{S_2} = \phi_{\{1,5,7\}}$

(e) $\mathfrak{L}(\{1,2\}) = -1.24$ (f) $\mathfrak{L}(\{1,3\}) = -0.39$ (g) Subset $\phi_{\{1,2,5\}}$ by LLR (h) $r_k$ vs. $\phi_k$

Figure 3: First row: Chloromethane dataset; second row: SDSS dataset in (e), (f) and (g), (h) show the example when LLR failed. (c) and (d) are embeddings with top two ranked subsets $S_1$ and $S_2$, colored by the distances between C and two different $Cl^-$, respectively. (e) and (f) are embeddings of $\phi_{\{1,2\}}$ (suboptimal set) and $\phi_{\{1,3\}}$ (maximizer of $\mathfrak{L}$), respectively (values shown in caption).

**Synthetic manifolds** The results of synthetic manifolds are in Figure 2. (i) Manifold with $s = d$. The first synthetic dataset we considered, $\mathcal{D}_1$, is a two dimensional strip with aspect ratio $W/H = 2\pi$. Left panel of the top row shows the scatter plot of such dataset. From the theoretical analysis in Section 3, the coordinate set that corresponds to slowest varying unique eigendirection is $S = \{1, \lceil W/H \rceil\} = \{1, 7\}$. Middle panel, with $S_* = \{1, 7\}$ selected by INDEIGENSEARCH with $\zeta$ chosen by REGUPARAMSEARCH, confirms this. The right panel shows the box plot of $\{\mathfrak{D}(S, i)\}_{i=1}^n$. According to the proposed procedure, we eliminate $S_0 = \{1, 2\}$ since $\mathfrak{D}(S_0, i) \geq 0$ for almost all the points. (ii) Manifold with $s > d$. The second data $\mathcal{D}_7$ is displayed in the left panel of the second row. Due to the mechanism we used to generate the data, the resultant torus is non-uniformly distributed along the z axis. Middle panel is the embedding of the optimal coordinate set $S_* = \{1, 4, 5\}$ selected by INDEIGENSEARCH. Note that the middle region (in red) is indeed a two dimensional narrow tube when zoomed in. The right panel indicates that both $\{1, 2, 3\}$ and $\{1, 2, 4\}$ (median

is around zero) should be removed. The optimal regularization parameter is $\zeta_* \approx 7$. The result of the third dataset $\mathcal{D}_{13}$, *three torus*, is in the third row of the figure. We displayed only projections of the penultimate and the last coordinate of original data $\mathbf{X}$ and embedding $\phi_{S_*}$ (which is $\{5, 10\}$) colored by $\alpha_1$ of (S15) in the left and middle panel to conserve space. A full combinations of coordinates can be found in Figure S5. The right panel implies one should eliminate the set $\{1, 2, 3, 4\}$ and $\{1, 2, 3, 5\}$ since both of them have more than 75% of the points such that $\mathfrak{D}(S, i) \geq 0$. The first remaining subset is $\{1, 2, 5, 10\}$, which yields an optimal regularization parameter $\zeta_* \approx 5$.

**Molecular dynamics dataset [FTP16]**    The dataset has size $n \approx 30,000$ and ambient dimension $D = 40$, with the intrinsic dimension estimate be $\hat{d} = 2$ (see Supplement H.1 for details). The embedding with coordinate set $S = [3]$ is shown in Figure 3a. The first three eigenvectors parameterize the same directions, which yields a one dimensional manifold in the figure. Top view ($S = [2]$) of the figure is a u-shaped structure similar to the yellow curve in Figure 1a. The heat map of $\mathfrak{L}(\{1, i, j\})$ for different combinations of coordinates in Figure 3b confirms that $\mathfrak{L}$ for $S = [3]$ is low and that $\phi_1$, $\phi_2$ and $\phi_3$ give a low rank mapping. The heat map also shows high $\mathfrak{L}$ values for $S_1 = \{1, 4, 6\}$ or $S_2 = \{1, 5, 7\}$, which correspond to the top two ranked subsets. The embeddings with $S_1, S_2$ are in Figures 3c and 3d, respectively. In this case, we obtain two optimal $S$ sets due to the data symmetry.

**Galaxy spectra from the Sloan Digital Sky survey (SDSS)**    [3] [AAMA$^+$09], preprocessed as in [MMVZ16]. We display a sample of $n = 50,000$ points from the first 0.3 million points which correspond to closer galaxies. Figures 3e and 3f show that the first two coordinates are almost dependent; the embedding with $S_* = \{1, 3\}$ is selected by INDEIGENSEARCH with $d = 2$. Both plots are colored by the blue spectrum magnitude, which is correlated to the number of young stars in the galaxy, showing that this galaxy property varies smoothly and non-linearly with $\phi_1, \phi_3$, but is not smooth w.r.t. $\phi_1, \phi_2$.

**Comparison with [DTCK18]**    The LLRCOORDSEARCH method outputs similar candidate coordinates as our proposed algorithm most of the time (see Table S3). However, the results differ for *high torus* as in Figure 3. Figure 3h is the leave one out (LOO) error $r_k$ versus coordinates. The coordinates chosen by LLRCOORDSEARCH was $S = \{1, 2, 5\}$, as in Figure 3g. The embedding is clearly shown to be suboptimal, for it failed to capture the cavity within the torus. This is because the algorithm searches in a sequential fashion; the noise eigenvector $\phi_2$ in this example appears before the signal eigenvectors e.g., $\phi_4$ and $\phi_5$.

**Additional experiments with real data**    are shown in Table 1. Not surprisingly, for most real data sets we examined, the independent coordinates are not the first $s$. They also show that the algorithm scales well and is robust to the noise present in real data.

Table 1: Results for other real datasets. Columns from left to right are sample size $n$, ambient dimension of data $D$, average degree of neighbor graph $\deg_{\mathrm{avg}}$, $(s, d)$ and runtime for IES, and the chosen set $S^*$, respectively. Last three datasets are from [CTS$^+$17].

|  | $n$ | $D$ | $\deg_{\mathrm{avg}}$ | $(s, d)$ | $t$ (sec) | $S_*$ |
|---|---|---|---|---|---|---|
| SDSS (full) | 298,511 | 3750 | 144.91 | (2, 2) | 106.05 | (1, 3) |
| Aspirin | 211,762 | 244 | 101.03 | (4, 3) | 85.11 | (1, 2, 3, 7) |
| Ethanol | 555,092 | 102 | 107.27 | (3, 2) | 233.16 | (1, 2, 4) |
| Malondialdehyde | 993,237 | 96 | 106.51 | (3, 2) | 459.53 | (1, 2, 3) |

The asymptotic runtime of LLRCOORDSEARCH has quadratic dependency on $n$, while for our algorithm is linear in $n$. Details of runtime analysis are Supplement F. LLRCOORDSEARCH was too slow to be tested on the four larger datasets (see also Figure S1).

# 7 Conclusion

Algorithms that use eigenvectors, such as DM, are among the most promising and well studied in ML. It is known since [GZKR08] that when the aspect ratio of a low dimensional manifold exceeds a threshold, the choice of eigenvectors becomes non-trivial, and that this threshold can be as low as 2. Our experimental results confirm the need to augment ML algorithms with IES methods in order to successfully apply ML to real world problems. Surprisingly, the IES problem has received little attention in the ML literature, to the extent that the difficulty and complexity of the problem have not been recognized. Our paper advances the state of the art by (i) introducing for the first time a differential geometric definition of the problem, (ii) highlighting geometric factors such as injectivity radius that, in addition to aspect ratio, influence the number of eigenfunctions needed for a smooth embedding, (iii) constructing selection criteria based on *intrinsic manifold quantities*, (iv) which have analyzable asymptotic limits, (v) can be computed efficiently, and (vi) are also robust to the noise present in real scientific data. The library of hard synthetic examples we constructed will be made available along with the python software implementation of our algorithms.

## Acknowledgements

The authors acknowledge partial support from the U.S. Department of Energy, Solar Energy Technology Office award DE-EE0008563 and from the NSF DMS PD 08-1269 and NSF IIS-0313339 awards. They are grateful to the Tkatchenko and Pfaendtner labs and in particular to Stefan Chmiela and Chris Fu for providing the molecular dynamics data and for many hours of brainstorming and advice.

## Footnotes

[1] In this paper, a smooth function or manifold will be assumed to be of class at least $\mathcal{C}^3$.

[2]We thank the anonymous reviewer who made this suggestion.

[3]The Sloan Digital Sky Survey data can be downloaded from `https://www.sdss.org`

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
