[Supplementary Material]

# Supplement to
## Selecting the independent coordinates of manifolds with large aspect ratios

## A  Notational table

Table S1: Notational table

| Matrix operation | |
|---|---|
| $\mathbf{M}$ | Matrix |
| $\mathbf{m}_i$ | Vector represents the $i$-th row of $\mathbf{M}$ |
| $\mathbf{m}_{:,j}^T$ | Vector represents the $j$-th column of $\mathbf{M}$ |
| $m_{ij}$ | Scalar represents $ij$-th element of $\mathbf{M}$ |
| $[\mathbf{M}]_{ij}$ | Scalar, alternative notation for $m_{ij}$ |
| $\mathbf{M}[\alpha,\beta]$ | Submatrix of $\mathbf{M}$ of index sets $\alpha,\beta$ |
| $\mathbf{v}$ | Column vector |
| $v_i$ | Scalar represents $i$-th element of vector $\mathbf{v}$ |
| $[\mathbf{v}]_i$ | Scalar, alternative notation for $v_i$ |
| **Scalars** | |
| $n$ | Number of samples |
| $D$ | Ambient dimension |
| $m$ | Dimension of diffusion embedding |
| $s$ | (Minimum) embedding dimension |
| $d$ | Intrinsic dimension |
| **Vectors & Matrices** | |
| $\mathbf{X}$ | Data matrix |
| $\mathbf{x}_i$ | Point $i$ in ambient space |
| $\mathbf{Y}$ | Diffusion coordinates |
| $\mathbf{y}_i$ | Point $i$ in diffusion coordinates |
| $\phi_i$ | The $i$-th diffusion coordinate of all points |
| $\mathbf{K}$ | Kernel (similarity) matrix |
| $\mathbf{L}$ | Graph Laplacian |
| $\mathbf{H}(i)$ | Dual metric at point $i$ |
| $\mathbf{I}_k$ | Identity matrix in $k$ dimension space |
| $\mathbf{1}_n$ | All one vector $\in \mathbb{R}^n$ |
| $\mathbf{1}_S$ | $[\mathbf{1}_S]_i = 1$ if $i \in S$ 0 otherwise |
| **Miscellaneous** | |
| $G(V,E)$ | Graph with vertex set $V$ and edge set $E$ |
| $\mathcal{M}$ | Data manifold |
| $\phi(\cdot)$ | Embedding mapping |
| $\mathfrak{L}(S;\zeta)$ | Utilities |
| $\mathfrak{R}$ | Unpenalized utilities |
| $[s]$ | Set $\{1,\cdots,s\}$ |
| $D(\cdot\|\cdot)$ | KL divergence |
| $\mathbf{D}$ | Jacobian |
| $\mathfrak{D}(S,i)$ | Leave-one-out regret of point $i$ |

## B  Pseudocodes

---
**Algorithm S1:** DIFFMAP

---
**Input** : Data matrix $\mathbf{X} \in \mathbb{R}^{n \times D}$, bandwidth $\varepsilon$,
embedding dimension $m$

1 Compute similarity matrix $\mathbf{K}$ with
$$K_{ij} = \begin{cases} \exp\left[-\frac{||\mathbf{x}_i-\mathbf{x}_j||^2}{\varepsilon^2}\right] & \text{if } \|x-y\| \le 3\varepsilon \\ 0 & \text{otherwise} \end{cases}$$

2 $\mathbf{L} \leftarrow \text{LAPLACIAN}(\mathbf{K}) \in \mathbb{R}^{n \times n}$ (Algorithm S2)

3 Compute eigenvectors of $\mathbf{L}$ for smallest $m+1$
eigenvalues $[\phi_0\,\phi_1\,\ldots\,\phi_m] \in \mathbb{R}^{n\times(m+1)}$
**Return:** $\mathbf{\Phi} = [\phi_1\,\ldots\,\phi_m] \in \mathbb{R}^{n\times m}$ The
*embedding coordinates* of $\mathbf{x}_i$ are
$(\Phi_{i1},\ldots,\Phi_{im}) \in \mathbb{R}^m$

---
**Algorithm S2:** LAPLACIAN

---
**Input** : Symmetric similarity matrix $\mathbf{K}$

1 Calculate the *degree* of node $i$,
$[\mathbf{w}]_i = \sum_{j=1}^n K_{ij} \triangleright \text{ Set } \mathbf{W} = \text{diag}(\mathbf{w})$

2 $\tilde{\mathbf{L}} = \mathbf{W}^{-1}\mathbf{K}\mathbf{W}^{-1}$

3 $[\tilde{\mathbf{w}}]_i \leftarrow \sum_{j=1}^n \tilde{L}_{ij} \triangleright \text{ Set } \tilde{\mathbf{W}} = \text{diag}(\tilde{\mathbf{w}})$

4 $\mathbf{L} = \mathbf{I}_n - \tilde{\mathbf{W}}^{-1}\tilde{\mathbf{L}}$
**Return:** Renormalized graph Laplacian $\mathbf{L}$

---
**Algorithm S3:** LLRCOORDSEARCH

---
**Input** : Embedding
$\mathbf{Y} = [\phi_1,\cdots,\phi_m] \in \mathbb{R}^{n\times m}$

1 Set the leave-one-out validation error
$\mathbf{r} = [1,\cdots,1] \in \mathbb{R}^m$

2 **for** $s = 2 \to m$ **do**

3     Bandwidth of LLR:
    $h \leftarrow \frac{1}{3}\cdot\text{MEDIAN}(\text{PAIRWISEDIST}(\phi_{[s-1]}))$

4     $\hat{\phi}_s \leftarrow$
    $\text{LOCALLINEARREGRESSION}(\phi_s, \phi_{[s-1]}, h)$

5     $r_s = \sqrt{\frac{\|\hat{\phi}_s-\phi_s\|^2}{\|\phi_s\|^2}}$

6 **end**

7 $S_* \leftarrow \text{ARGSORT}(\mathbf{r})$
  $\triangleright$ Sort in descending order.
**Return:** Sorted independent coordinates $S_*$

# C  Proofs and extra theorems

## C.1  Proof of Lemma 2

**Proof.**  Let $\mu^*_{\phi_S(\mathcal{M})}$ denote the Riemannian measure induced by $g_{*\phi_S}$. Since $(\mathcal{M}, g)$ and $(\phi_S(\mathcal{M}), g_{*\phi_S})$ are isometric by definition, $\int_\mathcal{M} f(\mathbf{x})d\mu_\mathcal{M}(\mathbf{x}) = \int_{\phi_S(\mathcal{M})} f(\phi_S^{-1}(\mathbf{y}))d\mu^*_{\phi_S(\mathcal{M})}(\mathbf{y}) = \int_{\phi_S(\mathcal{M})} f(\phi_S^{-1}(\mathbf{y}))\sqrt{\det g_{*\phi_S}(\mathbf{y})}d\mu_{\phi_S(\mathcal{M})}(\mathbf{y})$ follows from the change of variable formula. It remains to find the expression of $j_S(\mathbf{y}) = \sqrt{\det g_{*\phi_S}(\mathbf{y})}$. The matrix $\mathbf{U}_S(\mathbf{y})$ (note that $\mathbf{U}_S(\mathbf{y})$ is *not orthogonal*) can be written as

$$\mathbf{U}_S(\mathbf{y}) = \mathbf{V}\mathbf{Q}_S(\mathbf{y}) \tag{S1}$$

where $\mathbf{V} \in \mathbb{R}^{s\times d}$ is an orthogonal matrix and $\mathbf{Q}_S(\mathbf{y}) \in \mathbb{R}^{d\times d}$ is upper triangular. Then,

$$\mathbf{H}_S(\mathbf{y}) = \mathbf{U}_S(\mathbf{y})\mathbf{\Sigma}(\mathbf{y})\mathbf{U}_S(y)^\top = \mathbf{V}_S(y) \underbrace{\left(\mathbf{Q}_S(\mathbf{y})\mathbf{\Sigma}(\mathbf{y})\mathbf{Q}_S(\mathbf{y})^\top\right)}_{\tilde{\mathbf{H}}_S(\mathbf{y})} \mathbf{V}_S(\mathbf{y})^\top. \tag{S2}$$

In the above $\tilde{\mathbf{H}}_S(y)$ is the co-metric expressed in the new coordinate system induced by $\mathbf{V}_S(\mathbf{y})$. Hence, in the same basis, $g_{*\phi_S}$ is expressed by

$$\tilde{\mathbf{G}}_S(y) = \tilde{\mathbf{H}}_S(y)^{-1} = \left(\mathbf{Q}_S(\mathbf{y})\mathbf{\Sigma}(\mathbf{y})\mathbf{Q}_S(\mathbf{y})^\top\right)^{-1}. \tag{S3}$$

The volume element, which is invariant to the chosen coordinate system, is

$$\det\left(\mathbf{Q}_S(\mathbf{y})\mathbf{\Sigma}(\mathbf{y})\mathbf{Q}_S(\mathbf{y})^\top\right)^{-1/2} = \prod_{k=1}^d \sigma_k(\mathbf{y})^{-1/2}q_{S,kk}(\mathbf{y})^{-1}. \tag{S4}$$

From (S1), it follows also that

$$\det\left(\mathbf{Q}_S(\mathbf{y})\mathbf{\Sigma}(\mathbf{y})\mathbf{Q}_S(\mathbf{y})^\top\right)^{-1/2} = 1/\operatorname{Vol}\left(\mathbf{U}_S(\mathbf{y})\mathbf{\Sigma}(\mathbf{y})^{1/2}\right) \tag{S5}$$

■

## C.2  Proof of Theorem 3

**Proof.**  Because $\phi_S$ is a smooth embedding, $j_S(\mathbf{y}) > 0$ on $\phi_S(\mathcal{M})$, and because $\mathcal{M}$ is compact, $\min_{\phi_S(\mathcal{M})} j_S(\mathbf{y}) > 0$. Similarly, noting that $\tilde{j}_S(\mathbf{y}) \geq \prod_{k=1}^d \sigma_k^{-1/2}(\mathbf{y})$, we conclude that $\tilde{j}_S(\mathbf{y})$ is also bounded away from 0 on $\mathcal{M}$. Therefore $\ln j_S(\mathbf{y})$ and $\ln \tilde{j}_S(\mathbf{y})$ are bounded, and the integral in the r.h.s. of (8) exists and has a finite value. Now,

$$\frac{1}{n}\sum_i \ln\mathfrak{R}(S, \mathbf{x}_i) \to \int_\mathcal{M} \ln\mathfrak{R}(S, \mathbf{x})p(\mathbf{x})d\mu_\mathcal{M}(\mathbf{x}) = \mathfrak{R}(S, \mathcal{M}). \tag{S6}$$

$$\int_\mathcal{M} \ln\mathfrak{R}(S, \mathbf{x})p(\mathbf{x})d\mu_\mathcal{M}(\mathbf{x}) \tag{S7}$$

$$= \int_{\phi_S(\mathcal{M})} \ln\mathfrak{R}(\phi_S^{-1}(\mathbf{y}))p(\phi_S^{-1}(\mathbf{y}))j_S(\mathbf{y})d\mu_{\phi_S(\mathcal{M})}(\mathbf{y})$$

$$= \int_{\phi_S(\mathcal{M})}\left[\frac{1}{2}\ln\frac{\operatorname{Vol}\left(\mathbf{U}_S^\top(\mathbf{y})\mathbf{U}_S(\mathbf{y})\right)}{\tilde{j}_S(\mathbf{y})} - \frac{p(\phi_S^{-1}(\mathbf{y})\prod_{k=1}^d \sigma_k^{1/2}(\mathbf{y})}{p(\phi_S^{-1}(\mathbf{y})\prod_{k=1}^d \sigma_k^{1/2}(\mathbf{y})}\right]p(\phi_S^{-1}(\mathbf{y}))j_S(\mathbf{y})d\mu_{\phi_S(\mathcal{M})}(\mathbf{y})$$

$$= \int_{\phi_S(\mathcal{M})} \ln\frac{j_S(\mathbf{y})p(\phi_S^{-1}(\mathbf{y})}{\tilde{j}_S(\mathbf{y})p(\phi_S^{-1}(\mathbf{y})}p(\phi_S^{-1}(\mathbf{y}))j_S(\mathbf{y})d\mu_{\phi_S(\mathcal{M})}(\mathbf{y}) = -D(pj_S\|p\tilde{j}_S) \tag{S8}$$

■

## C.3 Submodularity of the utility functions

**Theorem S1.** *For a rank $d$ tangent space matrix $\mathbf{U} \in \mathbb{R}^{m \times d}$, if any submatrix $\mathbf{U}_S$, with index set $S \subseteq [m]$ and $|S| = s \geq d$, is rank $d$, we have $\mathfrak{R}_1$ be a submodular set function.*

**Proof.** W.L.O.G, set $n = 1$, with slightly abuse of notation, let $\mathbf{U} = \mathbf{U}_{T \cup \{i\}} \in \mathbb{R}^{(|T|+1) \times d}$. The matrix can be written in the following form

$$\mathbf{U} = \begin{bmatrix} \mathbf{T} \\ \mathbf{a} \end{bmatrix} = \begin{bmatrix} \mathbf{S} \\ \mathbf{V} \\ \mathbf{a} \end{bmatrix} \in \mathbb{R}^{(|T|+1) \times d}$$

With $\mathbf{U}_S = \mathbf{S}$, $\mathbf{U}_T = \mathbf{T}$ and $U_{\{i\}} = \mathbf{a}$ for set $S \subseteq T \subseteq [m]$ and $i \in [m] \backslash T$. Here $\mathbf{a} \in \mathbb{R}^{1 \times d}$. By the definition of $\mathfrak{R}_1$ in (2), one has (ignoring the constants)

$$\mathfrak{R}_1(S) = \log \det(\mathbf{S}^\top \mathbf{S})$$

$$\mathfrak{R}_1(T) = \log \det (\mathbf{T}^\top \mathbf{T})$$

$$\mathfrak{R}_1(S \cap \{i\}) = \log \det \left( \begin{bmatrix} \mathbf{S} \\ \mathbf{a} \end{bmatrix}^\top \begin{bmatrix} \mathbf{S} \\ \mathbf{a} \end{bmatrix} \right)$$

$$\mathfrak{R}_1(T \cap \{i\}) = \log \det(\mathbf{U}^\top \mathbf{U})$$

Denote $\partial_i f(S) = f(S \cup \{i\}) - f(S)$ for some function $f$, we have

$$\partial_i \mathfrak{R}_1(S) = \log \det(\mathbf{S}^\top \mathbf{S} + \mathbf{a}^\top \mathbf{a}) - \log \det(\mathbf{S}^\top \mathbf{S})$$

$$\partial_i \mathfrak{R}_1(T) = \log \det(\mathbf{T}^\top \mathbf{T} + \mathbf{a}^\top \mathbf{a}) - \log \det(\mathbf{T}^\top \mathbf{T})$$

The full rank of any submatrices guarantees the positive definiteness of $\mathbf{S}^\top \mathbf{S}, \mathbf{T}^\top \mathbf{T}$, by matrix determinant lemma [Har98], we have

$$\det(\mathbf{S}^\top \mathbf{S} + \mathbf{a}^\top \mathbf{a}) = \det(\mathbf{S}^\top \mathbf{S}) \left( 1 + \mathbf{a}(\mathbf{S}^\top \mathbf{S})^{-1} \mathbf{a}^\top \right)$$

Therefore

$$\partial_i \mathfrak{R}_1(S) = 1 + \mathbf{a}(\mathbf{S}^\top \mathbf{S})^{-1} \mathbf{a}^\top$$

Similar equation holds for set $T$. Therefore,

$$\partial_i \mathfrak{R}_1(S) - \partial_i \mathfrak{R}_1(T) = \log \frac{1 + \mathbf{a}(\mathbf{S}^\top \mathbf{S})^{-1} \mathbf{a}^\top}{1 + \mathbf{a}(\mathbf{T}^\top \mathbf{T})^{-1} \mathbf{a}^\top}$$

Because $\mathbf{T}^\top \mathbf{T} \succeq \mathbf{S}^\top \mathbf{S}$, we have $(\mathbf{S}^\top \mathbf{S})^{-1} \succeq (\mathbf{T}^\top \mathbf{T})^{-1}$ [HHJ90], which implies $\partial_i \mathfrak{R}_1(S) - \partial_i \mathfrak{R}_1(T) \geq 0$ for all $S \subseteq T \subseteq [m]$ and $i \in [m] \backslash T$. This completes the proof. ∎

**Theorem S2.** *$\mathfrak{R}_2$ is a submodular set function.*

**Proof.** W.L.O.G, set $n, d = 1$. With slightly abuse of notation, let $\mathbf{u} \leftarrow \mathbf{u}_1(i)$ and $\mathbf{u}_S \leftarrow \mathbf{u}_1^S(i)$. For any set $S \subseteq T \subseteq [m]$ and $i \in [m] \backslash T$, we have

$$\partial_i \mathfrak{R}_2(S) = \mathfrak{R}_2(S \cap \{i\}) - \mathfrak{R}_2(S) = \log \frac{\sum_{k \in S} u_k^2 + u_i^2}{\sum_{k \in S} u_k^2} = \log \frac{\Sigma_S + u_i^2}{\Sigma_S}$$

$$\partial_i \mathfrak{R}_2(T) = \mathfrak{R}_2(T \cap \{i\}) - \mathfrak{R}_2(T) = \log \frac{\sum_{k \in T} u_k^2 + u_i^2}{\sum_{k \in T} u_k^2} = \log \frac{\Sigma_S + \Sigma_{T \backslash S} + u_i^2}{\Sigma_S + \Sigma_{T \backslash S}}$$

Where $\Sigma_S = \sum_{k \in S} u_k^2$. By definition, we have $\Sigma_S, \Sigma_{T \backslash S}, u_i^2 \geq 0$. Therefore,

$$\partial_i \mathfrak{R}_2(S) - \partial_i \mathfrak{R}_2(T) = \log \frac{(\Sigma_S + u_i^2) \cdot (\Sigma_S + \Sigma_{T \backslash S})}{\Sigma_S \cdot (\Sigma_S + \Sigma_{T \backslash S} + u_i^2)}$$

$$= \log \underbrace{\left[ \frac{\Sigma_S^2 + \Sigma_S (\Sigma_{T \backslash S} + u_i^2) + u_i^2 \Sigma_{T \backslash S}}{\Sigma_S^2 + \Sigma_S (\Sigma_{T \backslash S} + u_i^2)} \right]}_{\geq 1} \geq 0$$

Which completes the proof. ∎

## D Greedy search

---

**Algorithm S4:** GREEDYINDEIGENSEARCH

---

**Input** : Orthogonal basis $\{\mathbf{U}(i)\}_{i=1}^{n}$, eigenvalues $\boldsymbol{\lambda}$, intrinsic dimension $d$, regularization parameter $\zeta$

1   Solve $S_* \leftarrow \text{argmax}_{S\subseteq[m];|S|=d;1\in S}\ \mathfrak{L}(S;\zeta)$.

2   **for** $s = d+1 \rightarrow m$ **do**

3      $k_* = \text{argmax}_{k\in[m]\backslash S_*}\ \mathfrak{L}(S_* \cup \{k\};\zeta)$

4      $S_* \leftarrow S_* \cup \{k_*\} \vartriangleright$ `Record order`

5   **end**

**Return:** Independent coordinates $S_*$

---

Inspired by the greedy version of submodular maximization [NWF78], a greedy heuristic has been proposed, as in Algorithm S4. The algorithm starts from an observation that the optimal value of the $S' = \text{argmax}_{S;d\leq|S|<s}\ \mathfrak{L}(S;\zeta)$ will often time be a subset of the optimal $S_*$ of (3). Since the appropriate cardinality of the set $S$ is unknown, we can simply scan from $|S| = d$ to $m$. The order of the returned elements indicates the significance of the corresponding coordinate.

## E Pseudocode for selection of regularization parameter $\zeta$

---

**Algorithm S5:** REGUPARAMSEARCH

---

**Input** : Threshold parameter $\alpha$

1   **for** $\zeta = \zeta_{\max} \rightarrow 0$ **do**

     $\vartriangleright$ $\zeta_{\max}$ `should be sufficiently large such that` $S_*(\zeta_{\max}) = [s]$

2      $S \leftarrow S_*(\zeta)$; $S_* \leftarrow$ NULL; $\zeta'' \leftarrow$ NULL

3      **for** $i \in [n]$ **do**

4          $\mathfrak{D}(S,i) \leftarrow \mathfrak{R}(S_*^i;[n]\backslash\{i\}) - \mathfrak{R}(S;[n]\backslash\{i\})$ from equation (4)

5      **end**

6      **if** PERCENTILE$(\{\mathfrak{D}(S,i)\}_{i=1}^n, \alpha) \leq 0$ **and** $S_* = NULL$ **then**

7          Optimal set $S_* \leftarrow S$

8          $\zeta' \leftarrow \zeta \vartriangleright$ `First found a set that satisfies the criterion.`

9      **else if** $S_* \neq NULL$ **and** $S_* = S_*(\zeta)$ **then**

10         $\zeta'' \leftarrow \zeta \vartriangleright$ `Searching for` $\zeta''$

11     **else if** $S_* \neq NULL$ **and** $\zeta'' \neq NULL$ **and** $S_* \neq S_*(\zeta)$ **then**

12        $\zeta_* \leftarrow \frac{1}{2}(\zeta' + \zeta'')$

13        **break** $\vartriangleright$ `Leave the loop when found` $\zeta'' = \min_{\zeta\geq 0} S_*(\zeta') = S_*(\zeta)$

14      **else**

15         **continue**

16      **end**

17   **end**

**Return:** Optimal set $S_*$, optimal regularization parameter $\zeta_*$

---

## F Computational complexity analysis

### F.1 The proposed algorithms

For computation complexity analysis, we assume the embedding has already been obtained. Therefore, the computational complexity for building neighbor graph and solving the eigen-problem of graph Laplacian can be omitted. This is also the case for LLRCOORDSEARCH.

**Co-metrics and orthogonal basis**  According to [PM13], time complexity for computing $\mathbf{H}(i) \in \mathbb{R}^{m \times m} \, \forall \, i \in [n]$ is $\mathcal{O}(nm^2\delta)$, with $\delta$ be the average degree of the neighbor graph $G(V, E)$. In manifold learning, the graph will be sparse therefore $\delta \ll n$. Time complexity for obtaining principal space $\mathbf{U}(i)$ of point $i$ via SVD will be $\mathcal{O}(m^3)$. Total time complexity will be $\mathcal{O}(nm^2\delta + nm^3)$.

**Exact search**  Evaluating the utility $\mathfrak{L}$ for each point $i$ takes $\mathcal{O}(sd^2)$ in computing $\mathbf{U}_S(i)^\top \mathbf{U}_S(i)$, $\mathcal{O}(d^3)$ in evaluating the determinant of a $d \times d$ matrix. Normalization ($\mathfrak{R}_2$ term) takes $\mathcal{O}(ds)$. Exhaustive search over all the subset with cardinality $s$ takes $\mathcal{O}\left(\binom{m}{s}\right)$. The total computational complexity will therefore be $\mathcal{O}(nm^s(d^3 + d^2s) + nm^2\delta + nm^3) = \mathcal{O}(nm^{s+3} + nm^2\delta)$.

**Greedy algorithm**  First step of greedy algorithm includes solving $\operatorname{argmax}_{S \subseteq [m]; |S| = d} \mathfrak{L}(S, d)$, which takes $\mathcal{O}(nm^d d^3) = \mathcal{O}(nm^{d+3})$. Starting from $s = d + 1 \to m$, each step includes exhaustively search over $m - s$ candidates, with the time complexity of evaluating $\mathfrak{L}$ be $n(d^3 + d^2s)$. Putting things together, one has the second part of the greedy algorithm be

$$\sum_{s=d}^{m} n(m - s)(d^3 + d^2s) = \mathcal{O}(nm^5) \tag{S9}$$

The total computational complexity will therefore be $\mathcal{O}(n(m^{d+3} + m^5 + m^2\delta))$.

## F.2 Time complexity of [DTCK18] & discussion

The Algorithm LLRCOORDSEARCH is summarized in Algorithm S3. For searching over fixed coordinate $s$, the algorithm first build a kernel for local linear regression by constructing a neighbor graph, which takes $\mathcal{O}(n \log(n)s)^4$ using approximate nearest neighbor search. The $s$ dependency come from the dimension of the feature. For each point $i$, a ordinary least square (OLS) problem is solved, which results in $\mathcal{O}(n^2 s^2 + ns^3)$ time complexity. Searching from $s = 2 \to m$ will make the total time complexity be

Figure S1: Runtimes of different IES algorithms on two dimensional long strip. Purple, yellow and red curves correspond to INDEIGENSEARCH, GREEDYINDEIGENSEARCH and LLRCOORD- SEARCH algorithm, respectively.

$$\sum_{s=2}^{m} n^2 s^2 + ns^3 + ns \log n = \mathcal{O}(n^2 m^3 + nm^4) \tag{S10}$$

For a sparse graph, the overheads of the IN- DEIGENSEARCH and GREEDYINDEIGENSEARCH algorithms come from the enumeration of the subset $S$. Because of the linear dependency on the sample size $n$, the algorithm is tractable for small $s$ and $d$. However, LLRCOORDSEARCH has a quadratic dependency on sample size $n$, which is more computationally intensive for large sample size. For large $s$ and $d$, one can use the techniques in difference between submodular function optimization (e.g. [IB12]) as $\mathfrak{R}_1, \mathfrak{R}_2$ are both submodular set function from Theorems S1 and S2. An empirical runtime plot for different algorithms can be found in Figure S1. The runtime was evaluated on two dimensional long strip with $s = d = 2$ and was performed on a single desktop computer running Linux with 32GB RAM and a 8-Core 4.20GHz Intel® Core™ i7-7700K CPU.

## G  A heuristic to determine whether $s$ is sufficiently large

To determine whether the given $s$ is large enough, we proposed a heuristic by checking the histogram of the point-wise unpenalized utility $\exp\left(\mathfrak{R}(S, d)\right)$ ($\mathfrak{R} = \mathfrak{R}_1 - \mathfrak{R}_2$). The value is essentially the *normalized projected volume* of each point $i$ and is bounded between 0 and 1. Ideally, a perfect choice of cardinality $|S|$ will results in a concentration of mass in larger utility region. The heuristic

Figure S2: (a) Original data of $\mathcal{D}_4$, *swiss roll with hole* dataset. Embeddings with coordinate subset to be (b) $S = \{1, 8\}$, (c) $S = \{1, 10\}$, (f) $S = \{1, 8, 10\}$ and (g) $S = \{1, 11\}$ on $\mathcal{D}_4$. (e) Histogram of point-wise normalized projected volume on $\mathcal{D}_4$ for top two ranking of subsets (purple and yellow) and the union of two sets (red) obtain from INDEIGENSEARCH algorithm.

works as follow: first checks the histogram of unpenalized utility on the top few ranked subsets in terms of $\mathfrak{L}$. If the spikes in the small utility regions are witnessed in the histogram, taking the union of the subsets and inspecting the histogram of unpenalized utility on the combined set again. If the spike in small utility region decreases, one can conclude that a larger cardinality size $|S|$ is needed for such manifold.

We illustrate the idea on *swiss roll with hole* dataset in Figure S2a. Figure S2b is the optimal subset of coordinates $S_* = \{1, 8\}$ selected by the proposed algorithm that best parameterize the underlying manifold. Figure S2d suggested on should eliminate $S_0 = \{1, 2\}$ because $\mathfrak{D}(S_0, i) \geq 0$ for all the points. However, as shown in Figure S2b, though it has low frequency and having rank 2 for most of the places, might not be better for data analysis for the very thin arms in left side of the embedding. Figure S2e is the histograms of the point-wise unpenalized utility on different subsets. Purple and yellow curve correspond to the histogram of top two ranked subsets $S$ from INDEIGENSEARCH. Both curves show a concentration of masses in small utility region. The histogram of point-wise unpenalized utility on $\{1, 8, 10\}$, which is the union of the aforementioned two subsets, shows less concentration in the small utility region and implies that $|S| = 3$ might be a better choice for data analysis. Figure S2f shows the embedding with coordinate $S = \{1, 8, 10\}$. The resultant embedding is still a two dimensional strip, with some *twisting effects* occurred in certain regions. The thin arc in Figure S2b , turns out to be a collapsed two dimensional manifold via projection, as shown in the upper right part of Figure S2f and left part of Figure S2c. Here we have to restate that the embedding in Figure S2b, although is a *degenerated* embedding, is still the best set one can choose for $s = 2$ such that the embedding varies slowest and has rank 2. However, choosing $s = 3$ might be better for data analysis.

## H    Additional experiments & details of the used datasets

In this paper, a total of 13 different synthetic manifolds are considered. Table S2 summarized the synthetic manifolds constructed and its abbreviations (from $\mathcal{D}_1$ to $\mathcal{D}_{13}$). Embedding results for the synthetic manifolds are in Figures S3, S4 and S5. The ranking of the first few candidate sets $S$ from INDEIGENSEARCH, GREEDYINDEIGENSEARCHand LLRCOORDSEARCH can be found in Table S3. The table shows the optimal subsets return by three different algorithms are often time the same, with exception for $\mathcal{D}_7$ *high torus* as discussed in Section 6.

Table S2: Abbreviations for different synthetic manifolds in this paper. The abbreviation with asterisk represents such dataset is discussed in main manuscript.

| | Manifold with $s = d$ |
|---|---|
| $\mathcal{D}_1^*$ | Two dimensional strip (aspect ratio $2\pi$) |
| $\mathcal{D}_2$ | 2D strip with cavity (aspect ratio $2\pi$) |
| $\mathcal{D}_3$ | Swiss roll |
| $\mathcal{D}_4$ | Swiss roll with cavity |
| $\mathcal{D}_5$ | Gaussian manifold |
| $\mathcal{D}_6$ | Three dimensional cube |
| | Manifold with $s > d$ |
| $\mathcal{D}_7^*$ | High torus |
| $\mathcal{D}_8$ | Wide torus |
| $\mathcal{D}_9$ | z-asymmetrized high torus |
| $\mathcal{D}_{10}$ | x-asymmetrized high torus |
| $\mathcal{D}_{11}$ | z-asymmetrized wide torus |
| $\mathcal{D}_{12}$ | x-asymmetrized wide torus |
| $\mathcal{D}_{13}^*$ | Three-torus |

Table S3: Results returned from different algorithms on different synthetic datasets.

| | Exact search | | | | | Greedy rank | LLR rank |
|---|---|---|---|---|---|---|---|
| | 1 | 2 | 3 | 4 | 5 | | |
| $\mathcal{D}_1$ | [1, 7] | [1, 8] | [1, 9] | [1, 10] | [1, 12] | [1, 7, 6, 4, 3, 2, 5] | [1, 7, 14, 16, 11, 18, 6] |
| $\mathcal{D}_2$ | [1, 4] | [1, 8] | [1, 9] | [1, 10] | [1, 12] | [1, 4, 8, 6, 5, 3, 2] | [1, 4, 8, 5, 17, 11, 14] |
| $\mathcal{D}_3$ | [1, 9] | [1, 10] | [1, 11] | [1, 13] | [1, 18] | [1, 9, 5, 2, 3, 4, 6] | [1, 9, 19, 16, 12, 10, 4] |
| $\mathcal{D}_4$ | [1, 8] | [1, 10] | [1, 11] | [1, 14] | [1, 15] | [1, 8, 3, 2, 4, 10, 5] | [1, 8, 11, 10, 19, 16, 4] |
| $\mathcal{D}_5$ | [1, 6] | [1, 8] | [1, 10] | [1, 11] | [1, 13] | [1, 6, 2, 8, 3, 10, 4] | [1, 6, 19, 8, 18, 14, 12] |
| $\mathcal{D}_6$ | [1, 2, 8] | [1, 2, 11] | [1, 4, 8] | [1, 2, 17] | [1, 2, 13] | [1, 2, 8, 3, 4, 6, 5] | [1, 2, 8, 10, 3, 13, 6] |
| $\mathcal{D}_7$ | [1, 4, 5] | [1, 4, 8] | [1, 5, 7] | [1, 7, 12] | [1, 7, 8] | [1, 5, 4, 3, 6, 2, 8] | [1, 2, 5, 4, 15, 6, 10] |
| $\mathcal{D}_8$ | [1, 2, 7] | [1, 4, 7] | [1, 3, 7] | [1, 2, 9] | [1, 5, 7] | [1, 7, 2, 4, 3, 13, 5] | [1, 2, 7, 13, 12, 15, 14] |
| $\mathcal{D}_9$ | [1, 3, 4] | [1, 3, 7] | [1, 4, 6] | [1, 3, 10] | [1, 7, 9] | [1, 3, 4, 2, 9, 7, 6] | [1, 3, 4, 2, 19, 8, 7] |
| $\mathcal{D}_{10}$ | [1, 2, 4] | [1, 3, 4] | [1, 4, 5] | [1, 6, 9] | [1, 6, 14] | [1, 4, 2, 3, 5, 6, 8] | [1, 4, 2, 3, 8, 5, 6] |
| $\mathcal{D}_{11}$ | [1, 2, 5] | [1, 4, 8] | [1, 4, 5] | [1, 8, 9] | [1, 2, 8] | [1, 5, 2, 4, 8, 3, 9] | [1, 2, 5, 8, 10, 9, 11] |
| $\mathcal{D}_{12}$ | [1, 2, 5] | [1, 4, 5] | [1, 2, 7] | [1, 3, 5] | [1, 2, 8] | [1, 5, 2, 3, 4, 6, 8] | [1, 5, 2, 6, 10, 9, 4] |
| $\mathcal{D}_{13}$ | [1, 2, 5, 10] | [1, 3, 5, 10] | [1, 4, 5, 10] | [1, 5, 6, 10] | [1, 2, 8, 10] | [1, 5, 10, 2, 4, 3, 6] | [1, 2, 10, 5, 14, 15, 16] |

## H.1 Description of Chloromethane dataset

Before we started, we put a detail of the chloromethane dataset used in Section 6 in the main text. In SN2 reaction molecular dynamics of chloromethane [FTP16] dataset, two chloride atoms substitute with each other in different configurations/points $\mathbf{x}_i$ as described in the following chemical equation $CH_3Cl + Cl^- \longleftrightarrow CH_3Cl + Cl^-$. The dataset exhibits some kind of clustering structure with a sparse connection between two clusters which represents the time when the substitution happened.

## H.2 Additional experiments on synthetic manifolds with $s = d$

Below summarized the details of generating the datasets.

1. $\mathcal{D}_1^*$: points from this dataset are sampled uniformly from $\mathbf{x}_i \sim \text{UNIF}([-2, 2] \times [-4\pi, 4\pi])$.

2. $\mathcal{D}_2$: points are first sampled uniformly from $[-2, 2] \times [-4\pi, 4\pi]$. Points $i$ are removed if $|X_{i1}| < 4\pi/3$ and $|X_{i2}| < 2/3$.

3. $\mathcal{D}_3$: first sampling points $\mathbf{X}_{\text{true}} = [\mathbf{x}_0, \mathbf{y}_0]$ uniformly from a two dimensional strip. The data $\mathbf{X}$ can be obtained by the following non-linear transformation.

$$\mathbf{X} = \left[ \frac{\mathbf{x}_0 \circ \cos \mathbf{x}_0}{2}, \mathbf{y}_0, \frac{\mathbf{x}_0 \circ \sin \mathbf{x}_0}{2} \right] \tag{S11}$$

With $\circ$ denotes Hadamard (element-wise) product.

4. $\mathcal{D}_4$: sampling points $\mathbf{X}_{\text{true}} = [\mathbf{x}_0, \mathbf{y}_0]$ uniformly from 2D strip with cavity then applying the transformation (S11) to get $\mathbf{X}$.

5. $\mathcal{D}_5$: sampling points $\mathbf{X}_{\text{true}}$ uniformly from ellipse $\left\{ (x, y) \in \mathbb{R}^2 : \left( \frac{x}{6} \right)^2 + \left( \frac{y}{2} \right)^2 = 1 \right\}$. The data is obtained by

$$\mathbf{X} = [\mathbf{X}_{\text{true}}, \mathbf{z}]$$

With $z_i = \exp \left( - \left( \left( \frac{X_{i1}}{3} \right)^2 + X_{i2}^2 \right) / 2 \right)$

6. $\mathcal{D}_6$: points are sampled uniformly from $[-1, 1] \times [-2, 2] \times [-4, 4]$.

The experimental results are in Figure S3 ($\mathcal{D}_4$ in Figure S2).

## H.3 Additional experiments on synthetic manifolds with $s > d$

### H.3.1 Tori and asymmetrized tori

A torus can be parametrized by

$$\begin{aligned} x &= (a + b \cos \alpha) \cos \beta \\ y &= (a + b \cos \alpha) \sin \beta \\ z &= h \sin(\beta) \end{aligned} \tag{S12}$$

1. $\mathcal{D}_7^*$: sampling $\boldsymbol{\alpha}, \boldsymbol{\beta}$ uniformly from $[0, 2\pi)$ and generating the torus with $(a, b, h) = (3, 2, 8)$ from (S12).

2. $\mathcal{D}_8$: generating the torus with $(a, b, h) = (10, 2, 2)$.

3. $\mathcal{D}_9$: generating a high torus with $(a, b, h) = (3, 2, 8)$ and applying the following transformation

$$z \leftarrow (z - \min(z))^\gamma / \varsigma \tag{S13}$$

with $(\gamma, \varsigma) = (3, 1500)$

4. $\mathcal{D}_{10}$: generating a high torus with $(a, b, h) = (3, 2, 8)$ and applying the following transformation

$$x \leftarrow (x - \min(x))^\kappa / \eta \tag{S14}$$

with $(\kappa, \eta) = (2, 10)$

5. $\mathcal{D}_{11}$: generating a wide torus with $(a, b, h) = (10, 2, 2)$ and applying transformation (S13) with $(\gamma, \varsigma) = (3, 50)$.

6. $\mathcal{D}_{12}$: generating a wide torus with $(a, b, h) = (10, 2, 2)$ and applying transformation (S14) with $(\kappa, \eta) = (3, 1000)$.

The experimental results are in Figure S4.

### H.3.2 Three-torus

The parameterization of the three torus is

$$\begin{aligned} x_1 &= a_1 \sin \alpha_1 \\ x_2 &= (a_2 + a_1 \cos \alpha_1) \sin \alpha_2 \\ x_3 &= (a_3 + (a_2 + a_1 \cos \alpha_1) \cos \alpha_2) \sin \alpha_3 \\ x_4 &= (a_3 + (a_2 + a_1 \cos \alpha_1) \cos \alpha_2) \cos \alpha_3 \end{aligned} \tag{S15}$$

Figure S6: $M^2$ and $\hat{d}$ vs. ranking of $\mathcal{D}_{13}$

To generate $\mathcal{D}_{13}$, we sample $\boldsymbol{\alpha}_k$ uniformly from $[0, 2\pi)$ for $k \in [3]$ and apply the transformation (S15) with $(a_1, a_2, a_3) = (8, 2, 1)$. The sample size for this dataset is $n = 50,000$. The experimental result of three-torus can be found in Figure S5.

Figure S3: Synthetic manifolds with minimum embedding dimension $s$ equals intrinsic dimension $d$. Rows from top to bottom represent *two dimensional strip with cavity* (aspect ratio $W/H = 2\pi$), *swiss roll*, *gaussian manifold* and *three dimensional cube* dataset, respectively. Columns from left to right are the original data $\mathbf{X}$, embedding $\phi_{S_*}$ with optimal coordinate sets $S_*$ chosen by INDEIGENSEARCH and the regularization path, respectively.

## H.4 Verification of the chosen subsets on synthetic manifolds

Unlike 2D strip, the close form solution of the optimal set is oftentimes unknown in general. In this section, we verify the correctness of the chosen subset by reporting the full procrustes distance (disparity score) $M^2$ [Dry16], which is defined to be the normalized sum of square of the point-wise difference between the procrustes transformed ground truth data $\mathbf{X}_{\text{true}} \in \mathbb{R}^{n \times k}$ and the test data $\mathbf{X}_{\text{test}} \in \mathbb{R}^{n \times k}$. Namely,

$$M^2(\mathbf{X}_{\text{true}}, \mathbf{X}_{\text{test}}) = \min_{\beta, \boldsymbol{\gamma}, \boldsymbol{\Gamma}} \|\mathbf{X}_{\text{true}} - \beta \mathbf{X}_{\text{test}} \boldsymbol{\Gamma} - \mathbf{1}_n \boldsymbol{\gamma}^\top\|_F^2$$

$$\text{s.t. } \beta > 0, \boldsymbol{\gamma} \in \mathbb{R}^k, \Gamma \in SO(k)$$

(S16)

Here $\beta$ is a scale parameter, $\boldsymbol{\gamma}$ is the centering parameter and $\boldsymbol{\Gamma}$ is a $k \times k$ rotation matrix. We further require $\|\mathbf{X}_{\text{true}}\|_F = 1$ so that the disparity score will be between 0 and 1. Intuitively, one can expect the optimal choice of eigencoordinates $S_*$ will yield a small disparity score $M^2(\mathbf{X}_{\text{true}}, \phi_{S_*})$, with score increases as the coordinate set $S$ contains duplicate parameterizations or $\phi_S$ contains *knots, crossings*, etc. (e.g., Figure S8). Note that the score can only be calculated when the ground truth data $\mathbf{X}_{\text{true}}$ is available. For dataset without obtainable ground truth, one cannot proposed to report the disparity score of $\phi_S$ and the original data $\mathbf{X}$ as the proxy of $\mathbf{X}_{\text{true}}$, for $\mathbf{X}$ might not be a affine transformation of $\mathbf{X}_{\text{true}}$, e.g., Swiss roll. Besides, small $M^2$ given $\phi_S$ does not imply $S$ is optimal, which will be clear in the discussion of Figure S2g. Besides disparity scores, we will also report the estimated dimension $\hat{d}$. One can expect the estimated dimension for the optimal set $\dim(\phi_{S_*})$ will be close to the intrinsic dimension $d$, while the estimated dimension for sets containing duplicate parameterizations will be smaller than the intrinsic dimension. One cannot propose to use it as a criterion to choose the optimal set, for the suboptimal sets can also have estimated dimensions closed to the intrinsic dimension, e.g., Figure 3g. Throughout the experiment, the dimension estimation method by [LB05] is used for its ability to estimate dimension among all candidate subsets fairly fast.

Blue and red curves in Figure S7 and S6 show the disparity scores and estimated dimensions versus ranking of coordinate subsets for different synthetic manifolds, respectively. As expected, we have an increasing in $M^2$ and decreasing in $\hat{d}$ with respect to ranking. We first highlight that the set that produces the lowest disparity score is not necessarily optimal, although $S_*$ does yield a small disparity. This can be shown in the example of $\mathcal{D}_4$ *swiss roll with hole* dataset. Figure S2g is the embedding $\phi_{S_3}$ of $\mathcal{D}_4$, with $S_3$ is ranked third subset in terms of $\mathfrak{L}(S; \zeta)$, that minimizes the disparity score $M^2$ in $\mathcal{D}_4$ as shown in Figure S7d. This is because

Figure S8: Embedding that has crossing.

the embedding of the subset $S_3 = \{1, 11\}$ has larger area on the left, compared to Figure Figure S2b. This balances out the high disparity caused by the *flipped* region between two *knots* in the embedding $\phi_{S_3}$ when matched with $\mathbf{X}_{\text{true}}$. Since all the ranked first subset has low disparity compared to other subsets, we have higher confidence saying that the ranked 1st subset is indeed the optimal choice for the synthetic manifolds.

Figure S4: Synthetic manifolds with minimum embedding dimension $s$ greater than intrinsic dimension $d$. Rows from top to bottom represent *wide torus*, *z-asymmetrized high torus*, *x-asymmetrized high torus*, *z-asymmetrized wide torus* and *x-asymmetrized wide torus*, respectively. Columns from left to right are the original data $\mathbf{X}$, embedding $\phi_{S_*}$ with optimal coordinate sets $S_*$ chosen by INDEIGENSEARCH and the regularization path, respectively.

Figure S5: Experiment on *three-torus* dataset. (a) Original data **X** of three torus. (b) Embedding $\phi_{S_*}$ with optimal coordinate sets $S_*$ chosen by INDEIGENSEARCH. Rows for both (a) and (b) from top to bottom are embedding colored by the parameterization $(\alpha_1, \alpha_2, \alpha_3)$ in (S15), respectively.

(a) $\mathcal{D}_1$

(b) $\mathcal{D}_2$

(c) $\mathcal{D}_3$

(d) $\mathcal{D}_4$

(e) $\mathcal{D}_5$

(f) $\mathcal{D}_6$

(g) $\mathcal{D}_7$

(h) $\mathcal{D}_8$

(i) $\mathcal{D}_9$

(j) $\mathcal{D}_{10}$

(k) $\mathcal{D}_{11}$

(l) $\mathcal{D}_{12}$

Figure S7: Verification of the correctness of the chosen sets in synthetic manifolds.

## Footnotes

[4]This is a simplified lower bound, see [DS13] for details.