[Reviews · NeurIPS 2019]

Reviewer 1



The authors propose a criterion and method for selecting independent diffusion coordinates to capture the structure of a manifold with a large aspect ratio. The ideas presented in the paper are original, and the paper is clearly written, well organized and scientifically sound. The theoretical background and new analysis are provided in a clear and well-written form. The authors provide sufficient information to allow reproducibility of the method. Simulations are provided to support the success of the method, furthermore, the method is compared to an alternative approach. The paper indeed addresses a real problem in manifold learning, and the proposed method might be used by others in the future. I have a few minor concerns: the authors do not relate to: "Non-Redundant Spectral Dimensionality Reduction", Michaeli et al. probably unintentionally. However, I believe that this method provides a true alternative to the proposed method and this should be addressed. -The choice of the kernel bandwidth ($\epsilon$) is not addressed, this parameter could dramatically affect the results. Moreover, in some cases, if $\epsilon$ is chosen as a diag matrix (i.e. different number for each coordinate), the aspect ratio problem could be fixed (see for example "Kernel Scaling for Manifold Learning and Classification"). To summarize, I think the paper should be accepted and hope that these minor changes could be easily addressed to improve this manuscript. Respond to rebuttal: The authors have addressed all my comments in the rebuttal, my opinion is unchanged, I think that the paper should be accepted with the appropriate edits included in the final version.

Reviewer 2



The authors provide a novel solution to the problem first identified in [DTCK18], that of identifying a parsimonious subset of eigenvectors from a diffusion map embedding. From the perspective of differential geometry, the authors identify a new criterion for evaluating the independence of a set of eigenvectors and use this to identify suitable independent subsets of eigenvectors of the diffusion map. This area of manifold embedding is relatively understudied, and the solution by the authors seems elegant, improves on existing work, and is scalable to large datasets. The paper is also accompanied by an impressive number of experiments. Comments: 1. The authors claim that their method is robust to "noise present in real scientific data". However, it is hard to determine whether or not this is the case given the examples provided. An experiment on synthetic data with added noise would improve this claim. 2. Some of the figures in the main text were difficult to parse. It appears that in Figure 1a the y-axis is mislabeled and contains multiple overlaid plots. It is also difficult to assess the utility of the embeddings provided for the real datasets in Figure 3 as there is no ground truth geometry that we can reference. It would be useful to know how the coloring used for the Chloromethane dataset (or what the data actually is) and to have some more interpretation of the utility of the embedding. Typographical comments: 1. Line 98: "regreesion" -> regression 2. "Chloromethane" is misspelled in the Fig. 3 legend Based on the strength of the experimental results and theoretical interpretation of this problem I recommend accepting this paper. Update (Aug 11): Reading the other reviews and the author feedback, my opinion of this paper has not changed. I agree with author three that considering the problem of selecting the idea subspace for conditionality reduction (as opposed to the ideal subset of eigenvector) is an interesting problem and perhaps will yield interesting progress in the field. However, that does not detract from the significance of the problem considered in this work. The authors are thorough and their response to request for analysis of robustness to noise is satisfactory. I do not wish to revise my score.

Reviewer 3



This paper studies the problem of selecting coordinates of a map into a high-dimensional Euclidean space (assumed to be a smooth embedding) to produce a smooth immersion into a lower-dimensional Euclidean space. As the original map is composed of the eigenfunctions of a Laplacian, the authors call this the Independent Eigencoordinate Selection problem. The main contribution of the paper is to design an objective function to encourage the projected map to be locally injective and a regularization term encouraging use of slowly-varying lower eigenvalues. The IES problem is naturally phrased as a subset selection problem given these choices. The paper does not focus on how to optimize this objective function; rather, the authors study the behavior of the exact solution (found via exhaustive search of small subsets) under changes in the regularization parameter as a *regularization path*. I would have liked to see more discussion of the particular objective function chosen. Section 5 of the paper states that in the limit of infinitely many samples, the objective function converges to a K-L divergence between two Riemannian volume forms, one of them a pullback and the other cooked up to rescale the pullback. It seems like this limit is intended to motivate the choice of objective function. In that case, it would have been helpful to introduce it earlier and to discuss it more: e.g, why is K-L between these two volume forms a good way to encourage local injectivity. More fundamentally, the IES problem chooses a composition of the original map with a very specific Euclidean projection: a projection along coordinate axes. Searching over subsets of coordinates seems hard in general (this paper mainly uses exhaustive search). Why is it better to search among subsets of the coordinates than to search over all projections, which would be more amenable to continuous optimization techniques (e.g. manifold optimization on the Grassmannian)? I found the section on the regularization path and choosing $\zeta$ hard to follow. It seems to use notation introduced in the supplementary material without referring to it. Some of the mathematical terminology and notation in the paper is non-standard. For example, the pullback of a metric is normally denoted $\phi^*g$, not $g_{*\phi}$. The paper refers to the pushforward of the metric, which is really the pullback by the inverse map, $(\phi^{-1})^*g$. Of course this only makes sense where the inverse is well-defined. Similarly, the classification of functional dependencies/knots and crossings may be standard in machine learning, but as far as I know mathematicians would call these failures of local injectivity and failures of injectivity, respectively. A map that is locally (infinitesimally) injective but not necessarily globally injective is an immersion. It would be helpful to use this standard term as this is what the paper is seeking. In Section 5, some notation is used without being introduced. For example, I do not see where $p$ is defined, nor what $\sigma_k(y)$ is. The jacobian determinant is defined as the volume of a matrix, which seems like a typo. The characterization of the regularization parameter $\zeta$ is inconsistent. For example, section 5 states that "a smaller value for the regularization term encourages the use of slow varying coordinate functions." In fact, increasing $\zeta$ should put more emphasis on low-frequency modes. The paper states that "The rescaling of $\zeta$ [in equation(10)] in comparison with equation (2) aims to make $\zeta$ adimensional." But it is also stated that the objective function $\mathfrak{L}$ from equation (2) converges to that in equation (10). In that case, the scaling should be consistent between the two equations. If adimensionality is desirable, why not aim for that in the original definition of the objective function?

[Author Response · NeurIPS 2019]

Thanks for the VERY careful, responsible and competent reviews our paper has received! We will implement all
improvements recommended in the 3 reviews. Here we comment only on the more significant questions raised.

**Reviewer 1** *" relate to: "Non-Redundant Spectral Dimensionality Reduction", Michaeli et al."* Will do. Thanks for
pointing us to this reference. *" The choice of kernel bandwidth ($\varepsilon$) not addressed."* For the real data, $\varepsilon$ was optimized as
in [JMM17]. For the synthetic data, $\varepsilon$ was chosen heuristically; since, experiments were rerun using [JMM17] (see also
below). *"if $\varepsilon$ is chosen as a diag matrix..., the aspect ratio problem could be fixed (see for example "Kernel Scaling for*
*Manifold Learning and Classification"). To summarize, I think the paper should be accepted and hope that these minor*
*changes could be easily addressed to improve this manuscript."* We will discuss this reference in final paper.

**Reviewer 2** *"... experiment on synthetic data with added noise"* Experiment with the $6.28 \times 2$ strip data (be-
low, left): Gaussian noise with standard deviation $\sigma$ and ambient dimension $D = 3$ was added; for each $\sigma$,
the $\varepsilon$ selection algorithm [JMM17] was run, as well as the INDEIGENSEARCH algorithm for selecting the co-
ordinate for embedding in the top row, and intrinsic dimension estimation [LB04]. $\hat{d}$ measures the degrada-
tion of the manifold structure due to noise, and Corr the recovery of $h$ (shorter dimension in stripe). We see
that INDEIGENSEARCH degrades little even when $\hat{d} \approx 2.75$. Similar experiment on tall torus is below, right.

In the submission, $\sigma = 0.05$ and
the heuristic $\varepsilon$ was $0.25$ for stripe
and $1.5$ for tall torus. *"... more*
*interpretation of the utility of the*
*embedding."* For MD data, the
embeddings represent "slow mo-
tions" of the molecule (e.g., rota-
tions of one group w.r.t. another);
for galaxy spectra, it is interest-
ing to compare Fig. 3.f. with the
"HR diagram principal sequence",

where stars align in spectral/brightness space in 1D, according to their ages. For galaxies, age of star population is
also a feature, but the manifold is 2D. We now also have experiments with similar good results for UMAP embeddings
initialized by coordinate sets chosen by INDEIGENSEARCH.

**Reviewer 4** *" the paper does not focus on how to optimize this objective function"* In a longer paper, optimization
will receive more space. See also below, and Supplements C, D, E1. Note that for the current data sets, the run times for
[JMM17]/DiffMap/INDEIGENSEARCH are approximately in the ratio 30/3/1 (synthetic) and 100/10/1 (real).

*" the INDEIGENSEARCH problem chooses a composition of the original map with a very specific Euclidean projection:*
*a projection along coordinate axes. ... Why is [searchign over sets better] than to search over all projections, ([by]*
*e.g. manifold optimization on the Grassmannian)?"* **This is a super-interesting question for future work, and we**
**thank the Reviewer for raising it.** Presently, we can say that: the loss $\mathcal{L}(S)$ extends in a straightforward way to
the Grassmanian manifold; $\mathcal{L}(P)$, with $P$ a projection matrix, is a difference of convex functions, while the original
$\mathcal{L}(S)$ is a difference of *submodular functions* – see Supplement. **Computational aspects:** for small $s$ or $m$, there are
only $\sim 200$ $\mathcal{L}$ calculations; the search for $S$ is insignificant compared to computing the embedding (in particular, the
neighborhood graph and $\varepsilon$ search). When $m, s$ grow, the brute force INDEIGENSEARCH cost will grow exponentially.
The user has the choice between more advanced discrete optimization over $S$, based on submodularity, vs continuous
optimization over $P$, but of essentially the same function. A minor but nice advantage of searching over sets is that it
only requires the manifold learning toolbox; a practitioner needs not get tools (e.g. `manopt`) for optimization over the
Grassmanian manifold.
Mathematically, however, the question is deep and significant: can there be an advantage in using a linear combination
of eigenfunctions, instead of a subset? More specifically, for manifolds with small injectivity radius and large aspect
ratios, could it be that the required embedding dimension $s$ is smaller if we optimize over the Grassmanian and not over
discrete subsets of coordinates? We did not find any answers to this in the literature (so far).

*"... why is K-L between these two volume forms a good way to encourage local injectivity."* Local injectivity is by
definition tied to a volume form $j$ (sorry for yet another unusual notation); the only question is how do we "compare it
with 0". We compare it with its maximum $\tilde{j}_S$; then we integrate over the "inability to reach the max", which is exactly
what a K-L divergence does. Stretching it some, $pj_S$ is the "data" and $p\tilde{j}_S$ is the "model", and we are looking for a
view $S$ of the data that agrees with the model. Here $p$ is the density of the data sampled from a distribution on $\mathcal{M}$, see
also Assumption 2 in the manuscript.

**References**

[JMM17]  "Improved graph Laplacian via geometric self-consistency" by Joncas et al., NeurIPS 2017
[LB04]    "Maximum Likelihood Estimation of Intrinsic Dimension" by Levina and Bickel, NeurIPS 2004.


[Meta-Review · NeurIPS 2019]

Congratulations! Most reviewers agreed the work has a strong contribution and interest to the NeurIPS community. In your revision, please focus on addressing comments on the reviews, in particular the requests for added discussion, changed/carefully-defined notation from reviewer 4.